# Different disease inoculations cause common responses of the host immune system and prokaryotic component of the microbiome in *Acropora palmata*

**Benjamin D. Young**[1,2,3]*, **Stephanie M. Rosales**[2,3], **Ian C. Enochs**[3], **Graham Kolodziej**[2,3], **Nathan Formel**[4], **Amelia Moura**[5], **Gabrielle L. D'Alonso**[6], **Nikki Traylor-Knowles**[1]

1 Department of Marine Biology and Ecology, Rosenstiel School of Marine, Atmospheric and Earth Science, University of Miami, Miami, Florida, United States of America, 2 Cooperative Institute of Marine and Atmospheric Science, Rosenstiel School of Marine Atmospheric, and Earth Science, University of Miami, Miami, Florida, United States of America, 3 Atlantic Oceanographic and Meteorological Laboratory, National Oceanic and Atmospheric Administration, Miami, Florida, United States of America, 4 Biology Department, Woods Hole Oceanographic Institution, Woods Hole, Massachusetts, United States of America, 5 Coral Restoration Foundation, Tavernier, Florida, United States of America, 6 Mote Marine Laboratory, Summerland Key, Florida, United States of America

* benjamin.d.young@noaa.gov

**Data Availability Statement:** Raw reads and metadata (for transcriptomic and prokaryotic samples) are available in NCBI's SRA under

## Abstract

Reef-building corals contain a complex consortium of organisms, a holobiont, which responds dynamically to disease, making pathogen identification difficult. While coral transcriptomics and microbiome communities have previously been characterized, similarities and differences in their responses to different pathogenic sources has not yet been assessed. In this study, we inoculated four genets of the Caribbean branching coral *Acropora palmata* with a known coral pathogen (*Serratia marcescens*) and white band disease. We then characterized the coral's transcriptomic and prokaryotic microbiomes' (prokaryiome) responses to the disease inoculations, as well as how these responses were affected by a short-term heat stress prior to disease inoculation. We found strong commonality in both the transcriptomic and prokaryiomes responses, regardless of disease inoculation. Differences, however, were observed between inoculated corals that either remained healthy or developed active disease signs. Transcriptomic co-expression analysis identified that corals inoculated with disease increased gene expression of immune, wound healing, and fatty acid metabolic processes. Co-abundance analysis of the prokaryiome identified sets of both healthy-and-disease-state bacteria, while co-expression analysis of the prokaryiomes' inferred metagenomic function revealed infected corals' prokaryiomes shifted from free-living to biofilm states, as well as increasing metabolic processes. The short-term heat stress did not increase disease susceptibility for any of the four genets with any of the disease inoculations, and there was only a weak effect captured in the coral hosts' transcriptomic and prokaryiomes response. Genet identity, however, was a major driver of the transcriptomic variance, primarily due to differences in baseline immune gene expression. Despite genotypic differences in baseline gene expression, we have identified a common response for components of the coral holobiont to different disease inoculations. This work

PRJNA895002. All physiological data, as well as analysis scripts and bioinformatic scripts, are available at https://github.com/benyoung93/resposne_of_acropora_palmata_to_disease_and_heat.

**Funding:** ICE, Grant number 31252, NOAA Coral Reef Conservation Program. ICE provided experimental design expertise, lab resources, and manuscript comments and edits. NTK supported by NSF Grant 1951826 and Protect our Reefs Grant 2018-23. NTK helped plan the experiment, and provided direction on bioinformatic analysis, as well as manuscript comments and edits on drafts. The funders had no role in study design, data collection and analysis, decision to publish, or preparation of the manuscript.

**Competing interests:** The authors have declared that no competing interests exist.

has identified genes and prokaryiome members that can be focused on for future coral disease work, specifically, putative disease diagnostic tools.

## Introduction

Infectious diseases are ubiquitous in marine environments, affecting both prokaryotic and eukaryotic life [1]. In healthy ecosystems, low levels of disease can help structure biological communities and can be considered part of a balanced ecosystem by influencing food webs and maintaining stable populations [2–4]. On the other hand, disease outbreaks can also break down biological units, such as food webs, by causing trophic cascades [5,6]. This can heavily affect socioeconomic activities, such as fishing and tourism, by reducing biological diversity, biomass, and functions [7–9]. Understanding the underlying mechanics of disease in marine environments, and what can cause shifts from low to high prevalence, is therefore an imperative area of research.

Stony corals (Class Anthozoa, Order Scleractinia) are keystone marine species that generate the framework of tropical coral reefs. Corals, just like all other marine organisms, are affected by low chronic levels of disease, as well as large disruptive disease outbreaks [10,11]. The first reported coral disease was described in the 1970's [12], and there are now over 22 coral diseases described worldwide (reviewed in [13,14]). Despite research efforts over the last 40 years, identification of the causative pathogen(s) of coral diseases has been challenging [15,16]. Classical coral disease diagnostic methodologies have tried to meet the metrics of Koch's postulates because this is considered the necessary criteria for disease diagnosis [17]. However, Koch's postulates were developed primarily for terrestrial systems and, as such, fulfillment has not been possible for many coral diseases due to many marine prokaryotes being unknown and unculturable [18,19], causative agents being consortiums of microbes [20], or diseases being linked to abiotic factors [21–24]. Further complexity is added due to the coral organism. Corals are holobiont organisms with a complex microbiome that consists of members such as bacteria, archaea, viruses, fungi, and microeukaryotes [25]. Due to the complex assemblage of microbiome members in the coral holobiont, it has been difficult to discern between beneficial and detrimental organisms, with putative pathogens also commonly found in healthy coral microbiomes [13,16,26,27]. Within a species, genet identity is also an important variable, with differences in disease resistance and susceptibility commonly observed [28–30], but the genomic variables driving this have yet to be deduced. Advances in technology are now providing tools to overcome some of these challenges in coral disease research. Specifically, next-generation sequencing (NGS) and omics methodologies have allowed more in-depth characterization of the coral host and its microbiome. For the coral hosts, genomic analyses have revealed that corals have a complex immune system that covers the three key phases of innate immunity: Recognition, Signaling Pathways, and Effector Responses [31–34]. The immune processes that corals possess have also been shown to have high similarity to higher metazoans, indicating early evolutionary development of this system and conservation through time and space [33–35]. Transcriptomic work has taken this one step further, showing the coral's innate immune system plays a key role in response to pathogen stimulus and active disease signs (reviewed in [36,37–47]). Within the coral microbiome, sequencing of the 16S ribosomal RNA (rRNA) gene has advanced our understanding of the prokaryotic component of the microbiome (bacteria and archaea; hereafter referred to as the prokaryiome) in healthy and diseased corals. Healthy coral microbiomes have been shown to house stable prokaryiomes over time and

space [48]. Active disease signs and disease inoculation cause shifts away from these stable states, resulting in a pathobiome which, for the prokaryiome, is characterized by different microbial assemblages and accompanied by increases in species diversity [16,49–52]. Despite this, the exact mechanisms and causes of shifts in the prokaryiome are not well understood, with a wide range of potential mechanisms hypothesized [16]. While studies have looked at disease responses in corals, a comparison between two diseases, and how they affect the coral host and the prokaryiome, has not been extensively investigated. For the coral host, identification of which components of the innate immune system are activated to differing classes of pathogenic inoculations may provide an alternate diagnostic tool for future coral diseases. Similarly, identifying whether different pathogenic sources cause a common or unique secondary infection for the prokaryiome could provide potential species-wide treatments in the form of probiotics [53,54].

Coral disease has been a historically difficult research field, and climate change, specifically ocean warming, is providing additional complexities. For the coral host, increasing water temperatures cause corals to live closer to their thermal maxima which can negatively affect their physiological performance [55–57]. Thermal stress can also increase disease susceptibility due to energetically expensive functions such as the immune response not being initiated or maintained [34], and an increased risk of microbiome dysbiosis, secondary infection, and active disease signs [16,58,59]. Short-term heat stress (STHS) that doesn't reach the bleaching threshold (expulsion of the symbiotic algae) can influence coral health [60], but little is understood about how this affects disease susceptibility.

The Caribbean branching coral, *Acropora palmata*, is a candidate species to address these questions due to being heavily used in coral restoration [61–63], noted as more bleaching-tolerant [64–66], and, historically, disease-prone. Two well-documented *A. palmata* diseases are: white band type I (WBTi) [67–69] and Acroporid serratioses [70,71]. Since the 1970's, WBTi disease has caused widespread *A. palmata* population decline [67,68,72]. Despite this, the causative pathogen(s) are still unknown, with research identifying potential bacterial [73–75] and/or viral [76] organisms involved. On the other hand, Acroporid serratioses is caused by *Serratia marcescens* and is one of the few coral diseases to have a known causative pathogen [70,71].

The goal of this study was to investigate the physiological, transcriptomic, and prokaryiome responses to the known Acroporid serratioses pathogen (*S. marcescens*) and the unknown pathogen(s) of WBTi disease, and whether STHS would increase disease susceptibility as characterized by reduced immune function and prokaryiome dysbiosis. We hypothesized that the different pathogenic inoculations would elicit different transcriptomic responses in the coral host, as well as shifts to different states of the prokaryiome. We also hypothesized that the STHS would not cause increased disease susceptibility, but that there would be STHS signals present in the prokaryiome, and the coral hosts' transcriptomic response.

## Methods

### Collection, fragmentation, recovery

Roughly 32 fragments (11cm by 11cm) of four genets (HS1, CN2, ML2, and CN4) of *A. palmata* were collected in July 2019 from the Coral Restoration Foundation's (CRF) offshore nursery in Key Largo, Florida, and transported to four raceways in the Experimental Reef Lab (system described in [77]) in Miami, FL. Tank parameters were as follows for all aspects of the experiment unless specified; water temperature 27.5°C, max irradiance of 250 micromol m$^{-2}$ s$^{-1}$ between 10:00 and 14:00, two Koralia circulation and wave pumps (Hydor) to generate in tank water flow, and three nightly feedings resulting in 1.5g Reef Roids (Polyp Lab) distributed into each raceway. Tank water temperature was set to 27.5°C following temperature readings

at depth within the coral nursery, with this also set as the ambient temperature treatment. Following recovery, corals were trimmed to produce 6cm by 3cm fragments and affixed to acrylic plugs. Fragmentation yielded 40 fragments of each genet to be used in the experiment, with recovery from fragmentation occurring for three weeks.

## Short-term heat stress

Following fragmentation recovery, corals from the four genets were evenly and randomly allocated between 10 experimental tanks as described in [77]. Coral fragments were then subjected to one of two temperature treatments, with five tanks allocated to each: Ambient, 27.5°C for 15-days; short-term heat stress (STHS), 5-day temperature ramp from 27.5°C to 30°C (+0.5°C d$^{-1}$), 5-days at 30°C, 5-day ramp down to 27.5°C (-0.5°C d$^{-1}$). STHS was classified as 30°C due to reefs in the Florida Keys being predicted to bleach when average temperatures are >30°C for 16 days [66], and *A. palmata* showing increased bleaching incidence after 10-days at 31°C [78]. Therefore, the STHS at 30°C for five days was chosen as it was below the temperature and time thresholds identified for bleaching to occur. Allocation to temperature treatments resulted in 20 fragments of each genet receiving either the ambient, or STHS, temperature treatment.

## Disease inoculations and disease exposure

Following the 15-day temperature treatments, coral fragments were randomly allocated to individual jars in a custom disease inoculation setup housed within four raceways (S1A & S1B Fig). Each raceway was filled with circulating water and five racks. Each rack held eight jars, with each jar receiving its own seawater source. The top of each jar was raised above the raceway water level to allow outflow into the raceway to mitigate cross-contamination while increasing replication within a raceway. For each jar, the water turnover rate occurred every five minutes.

Two days after allocation to the disease inoculation setup, each coral fragment within its respective jar received two doses (Day-0 and Day-3) of one of four inoculations: white band type I disease slurry (WBTi DS), healthy tissue slurry (HTS), *S. marcescens* (*SM*), or *Serratia* placebo (*SP*). Triplicate samples of the WBTi DS, *SM* and *SP* were taken on day-0 and day-3, whereas triplicate samples for the HTS were only taken on day-0. Samples were saved in RNA-later (Thermofisher, MA) and stored at -80°C. For each temperature treatment (ambient or STHS), this resulted in five fragments of each genet receiving one of the four disease inoculations.

The WBTi DS was prepared from infected *A. cervicornis* fragments from multiple genets following the methodology used in [28]. Twelve diseased fragments of *A. cervicornis* were randomly chosen, and tissue within 5cm of the disease margin was removed using an airbrush filled with 0.2μm filtered autoclaved seawater (FAS). Removed tissue was pooled and diluted with additional 0.2μm FAS resulting in a concentration of 11cm$^2$ diseased tissue to 100 ml FAS. A total of 20 ml of the diluted slurry was added to each jar, using a 50 ml serological pipette [28]. The day-0 and day-3 inoculations were both prepared on their respective day using different fragments of diseased *A. cervicornis* fragments.

The HTS was prepared from multiple genets of *A. cervicornis* collected from the Rescue a Reef Key Biscayne Nursery in Miami, FL. Fragments were deemed healthy if there were no visible signs of stress such as bleaching, disease lesions, or predation marks. The HTS followed the same protocol as in [28]. Tissue was removed using an airbrush filled with 0.2μm FAS, pooled, and diluted with additional 0.2μm FAS to a concentration of 10cm$^2$ healthy tissue to 100 ml FAS. Like WBTi DS slurries, 20ml of HTS was added to its respective cohort. The same

healthy fragments of *A. cervicornis* were used to generate the Day-0 and Day-3 inoculations, with half of each fragment's tissue removed on day-0, and the other half removed on day-3.

A live culture of the PDR60 strain of *S. marcescens* (from [79]) was maintained on tryptic soy agar plates and inoculation followed [71]. A single colony from the tryptic soy agar cultures was added to a solution of 400 ml tryptic soy broth and 400g of sterile calcium carbonate powder. Following incubation (15hrs, room temperature), the bacterial broths were aliquoted into 50ml falcon tubes and centrifuged. The supernatant was removed leaving pelleted calcium carbonate powder with formed bacterial colonies [71]. For each falcon tube, 50ml of 0.2μm FAS was added to resuspend the pelleted calcium carbonate powder and bacterial colonies. To stay consistent with inoculation load from [71], 5ml of the suspended calcium carbonate with bacterial colonies was administered directly to live *A. palmata* fragments in their jars. Fresh *SM* inoculations were generated for the day-0 and day-3 inoculations.

The placebo *S. marcescens* inoculation followed the same protocol as the *S. marcescens* inoculation, without the addition of the cultured PDR60 strain [71]. The final calcium carbonate sterile 0.2μm filtered seawater solution therefore had no *S. marcescens*.

## Physiological measurements and analysis

During the recovery and temperature treatment portions of the experiment, Imaging Pulse Amplitude Modulation fluorometry (I-PAM) and fragment buoyant weight were used as health proxies for *A. palmata* fragments, see S1 File for methods and statistical analyses. Relative risk (RR) analyses compared the chances of disease signs occurring for different subsets of the *A. palmata* fragments. This included 1) temperature treatment for each genet within the WBTi DS and HTS, and within the *SM* and *SP*, 2) the four genets without temperature treatment included WBTi DS and HTS, and *SM* and *SP*, and 3) the four genets excluding temperature treatment and pooling disease inoculations (WBTi DS and *SM*) and control inoculations (HTS and *SP*).

## Tissue sampling, nucleic acid extractions, library prep, and sequencing

All tissue samples were roughly 0.5cm by 0.5cm, placed in 1.5ml of RNAlater (Thermofisher, MA) in 2ml cryovial tubes, and stored at -80˚C. All coral fragments were sampled the day before the first disease inoculation with these identified from hereon as pre-exposure (day-0) coral samples. The visual health status (VHS) of the coral fragments was recorded twice daily (~09:30 and 16:30) for 10 days. All coral fragments were assigned one of the two following metrics after day-0. "Visually healthy" was assigned to coral fragments not exhibiting visual signs of disease on day 10 (Fig 1A). "Diseased" was assigned to corals exhibiting any visual signs of disease between days 1–10 (Fig 1B). All visually healthy coral samples were taken on Day 10. For diseased coral fragments, samples were taken from Day 1 to Day 10 due to disease causing complete mortality in some coral fragments before the 10-day observation period was over.

DNA and RNA were extracted from the same piece of tissue for each coral sample using the MagBead RNA/DNA extraction kit (Zymo Research, Irvine, CA) and the Kingfisher Flex (Thermofisher, Waltham, MA). Please see S1 File, as well as [80], for full protocol and Kingfisher Flex scripts. For the coral host gene expression, the Quant-Seq FWD kit (Lexogen, Vienna) with qPCR add-on kit (for amplification cycle refinement) was utilized for 3'RNA-seq library preparation following the high yield/quality manufacturer protocol. RNA libraries were indexed with the add-on dual indexing kit (Lexogen, Vienna) and sent for sequencing at the Hussman Institute for Human Genomics (HIHG) at the Miller School of Medicine (University of Miami). Library quality control (QC) was performed by the sequencing facility (Tapestation

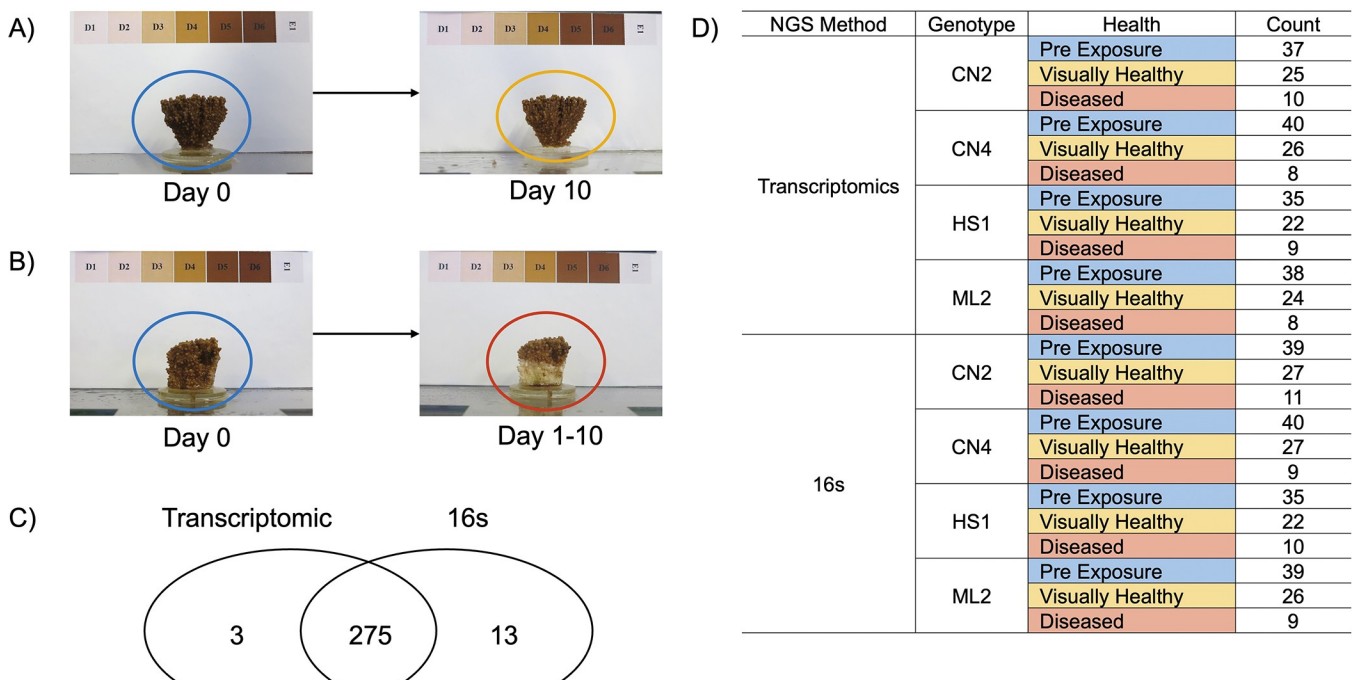

**Fig 1. Summary of sequenced transcriptomic, 16S rRNA samples, and visual designations of coral health.** A) Example of a pre-exposure coral (fragment in blue circle) before disease inoculation, and then after 10-days (fragment in yellow circle). Due to no visual signs of disease, fragments like this were classified as visually healthy for transcriptomic and 16s analysis. B) Example of a pre-exposure coral (fragment in blue circle) before disease inoculation, and then after 10-days showing visual signs of disease (fragment in dark red circle). C) Due to dual DNA and RNA extractions, there were 275 coral samples having transcriptomic and 16s data from the same piece of tissue. Three transcriptomic samples did not have complimentary 16s data available, and 13 16s rRNA samples did not have complimentary transcriptomic data due to failed sequencing or removal of low-quality samples during quality control. D) Table showing the breakdown of transcriptomic and 16s samples within each genet (column = Genet) and the identified visual health status [VHS] (column = Health). For the Health column, the color of the cell fill is consistent for all subsequent figures (pre-exposure = blue, visually healthy = yellow, diseased = dark red).

and Fluorometry) before pooling. Samples were then sequenced on a NovaSeq for single-end reads. For the coral prokaryiome, DNA was PCR amplified with the 16S rRNA gene V4 primers [81] using the Earth Microbiome Project (EMP) protocol [82–84]. Please see S1 File for the in-depth protocol. Pooled samples were sent to the HIHG at the Miller School of Medicine (University of Miami) for sequencing on two MiSeq runs using PE-300v3 kits.

## Coral host 3'RNA-seq analyses

All transcriptomic analyses pipelines and bioinformatic analyses are available at https://github.com/benyoung93/resposne_of_acropora_palmata_to_disease_and_heat. For the pre-processing pipeline (see S1 File), raw reads were aligned to the *A. palmata* reference genome and transcriptome [85,86], with downstream analysis using reads aligned to the reference transcriptome. Principal component analysis (PCA) was run using the variance stabilized transformed (VST) of raw filtered gene counts for different subsets of sequenced coral samples. Experimental variables were correlated to principal components (PC) using PCATools [87]. To identify finer scale patterns and remove batch or variance driving variables, limma [88] was used. Permutational multivariate analysis of variance (PERMANOVA) was run using vegan [89] on metadata variables of interest using the VST, and batch-removed VSTs as input. For multilevel factors, the initial PERMANOVA was run with a subsequent pairwise comparison [90].

To identify differences between the four genets (CN2, CN4, ML2, and HS1) the likelihood ratio test (LRT) methodology in DeSeq2 [91] was used with the full model (~*Temperature Exposure + Genet*) and reduced model (~ *Genet*). All significant genes identified from the LRT analysis (p-adj < 0.001) were used as input for Gene Ontology (GO) and Kyoto Encyclopedia of Genes and Genomes (KEGG) pathway enrichment analysis.

The VST counts, with genet variance removed, were used as input for a weighted gene correlation network analysis (WGCNA) using the R package wgcna [92]. The Ward2 method was used for initial clustering to identify any outlier samples. A single signed network with manual network construction was built with the following key parameters: soft power = 7, minimum module size = 30, deep split = 2, merged cut height = 0.25, cutHeight = 0.986. The eigengene of each module was correlated to metadata variables of interest and the most connected gene (hub gene) for each module was identified. Visualization of module-to-trait correlation and significance was done using Complex Heatmap [93]. All genes within modules were used as input for subsequent KEGG pathway and GO enrichment analyses following the methods below. Significant GO and KEGG pathways were used to infer each module's putative function.

GO enrichment analysis was run using Cytoscape v3.8.2 [94] and BiNGO v3.0.3 [95] using the GO to gene identifiers available in the *A. palmata* genome annotation file [85,86]. The hypergeometric test was utilized, and *p*-values were corrected with a Benjamini & Hochberg false discovery rate (FDR) with alpha set at 0.01. The genes remaining in the *A. palmata* genome after low count filtering were used as the background set for enrichment testing. Visualization of GO enrichment was undertaken in Cytoscape to identify relationships between significantly enriched child and parent terms. Genes present in significantly enriched terms were then visualized using the VST counts and Complex Heatmap [93].

KEGG enrichment was run using the gene to KEGG ID's available in the *A. palmata* genome annotation file [85,86] and the R package clusterprofiler v3.18.1 [96]. Enrichment was tested against the KEGG Orthology database (organism = ko) with alpha set at 0.01. Visualization of enriched pathways was undertaken in clusterprofiler [96], enrichPlot 1.10.2 [97], and GGPlot2 [98].

## Coral prokaryiome 16S rRNA analysis

For pre-processing of raw reads please see S1 File. Alpha diversity analysis (Shannon-Weiner) of the prokaryiome was run at the ASV level for each sample using the rust implementation (https://github.com/mooreryan/divnet-rs) of the R package divnet v0.3.7 [99], with subsequent significance testing using Breakaway v4.7.3 [100]. Shannon-Wiener estimates with confidence intervals were then used in Breakaway [101] to identify significant differences between Shannon-Wiener and metadata variables.

Relative abundance (RA) analysis was undertaken for different sample subsets and agglomerated to different taxonomic levels for variables of interest. Visualization of RA was undertaken in GGplot2 [98]. The threshold for retained taxa was greater than 10% unless specified. Raw filtered counts for all samples were transformed using a center logged ratio (CLR) transformation in CoDaSeq v0.99.6 [102,103]. The CLR-transformed counts were used as input in PCA with the lower 10% of variance removed and visualized using GGplot2 [98].

Vegan [89] was used for beta diversity analysis of the inoculation samples and subsets of the coral samples. Beta-diversity differences between groups was assessed for metadata variables of interest using the function adonis [89]. For multilevel factors, initial PERMANOVA was run with a subsequent pairwise comparison using the pairwiseAdonis package [90]. Alpha was set to 0.05 for PERMANOVA and post-pairwise comparisons.

CLR-transformed counts were used as input for a co-abundance analysis using wgcna [92]. The Ward2 method was used for initial clustering to identify sample outliers, before a single signed network with manual network construction was built with the following key parameters: cutHeight = 800, soft power = 12, minimum module size = 10, deep split = 2, merged cut height = 0.35. The eigen values from the ASVs were correlated to metadata variables of interest and the most connected ASVs (hub ASV) for each module were identified. Visualization of module-to-trait correlation and significance was done using Complex Heatmap [93].

The inferred metagenomic function (IMF) of the 16S rRNA abundance data was inferred using PICRUSt2 [104] using the filtered physeq object generated in pre-processing (see S1 File) for all coral samples. ASV sequences were placed into the PICRUSt2 reference tree with poorly aligned sequences removed. Hidden state predictions of gene families were inferred with the following trait options; 16s, EC and KO. Predictions were generated, and the previously calculated trait options and metagenomic functions were inferred. The IMF, using the KEGG orthology (KO) database, was used as input for co-expression analysis using wgcna [92]. The Ward2 method was used for initial clustering to identify sample outliers, before a single signed network with manual network construction was built with following key parameters: cutHeight = 8e-06, soft power = 8, minimum module size = 30, deep split = 2, merged cut height = 0.25. The eigengene was correlated to metadata variables of interest, with correlation and significance plotted using Complex Heatmap [93]. Modules were deemed significantly correlated at alpha 0.05. Hub genes were identified and predicted gene sets within modules were used for KEGG enrichment analyses using clusterprofiler [96] and the KEGG Orthology database (organism = ko) with alpha set to <0.05. Enriched KEGG terms were then used to assign putative module functions.

## Correlative analysis of the coral host gene expression to the prokaryiomes abundance and inferred metagenomic function

Integration of transcriptomic and 16S rRNA (abundance and IMF) WGCNA analyses were run using the module eigengenes from each analysis. The first module eigengene represents the first principal component of a module as an eigen vector, which therefore results in a value for each sample used in the analysis. Thus, these multi value vectors can be correlated against one another allowing correlative analysis between the different omics WGCNA analyses. Correlative analysis of module eigengenes was only run on samples that had complimentary transcriptomic and 16S rRNA data available. Correlations were calculated using the base R cor function [105], and correlation significance between transcriptomic and 16S rRNA modules was calculated using a student asymptotic $p$-value function in wgcna [92]. Visualization of correlation and significance was undertaken using Complex Heatmap [93].

# Results

## Transcriptomic and prokaryiome sequencing results

A total of 297 coral samples were successfully sequenced for transcriptomic analysis, with a median read depth of 4.4 million reads per sample (Fig 1D). The mean mapping rate was 28% to the *A. palmata* transcriptome, and 66% alignment to the *A. palmata* genome [85,86]. For downstream analysis, the samples aligned to the transcriptome were used. Fifteen samples with <1,000,000 reads were removed, resulting in 282 coral samples for downstream analysis. Of the 29,181 genes present in the *A. palmata* genome, 16,950 were retained after filtering of low count genes with less than two counts in greater than 20 samples. The 16,950 retained genes were used as the background gene set for GO enrichment tests. Raw reads and metadata are available in the NCBI's SRA under PRJNA895002.

A total of 315 samples were successfully sequenced for the prokaryiome, with this consisting of 300 coral samples (Fig 1D) and 15 disease inoculation samples. For the coral samples, six samples were removed due to low sequencing depth (<2,000 reads). For the remaining 294 samples, 9,182 ASVs were identified (S2 File), with 8,850 remaining after removal of 332 Chloroplast, Mitochondria and Eukaryotic sequences. Low count filtering (<10 ASVs across all samples) resulted in 5,564 ASVs for downstream analysis (S2 File). For the 15 disease inoculation samples, low abundance ASVs (<10 ASVs across all inoculation samples) were removed resulting in 445 ASVs for disease inoculation analysis (S2 File). Raw reads and metadata are available in the NCBI's SRA under PRJNA895002.

After quality filtering, 275 coral samples had corresponding transcriptomic and 16S rRNA data (Fig 1C). There were three samples that only had transcriptomic data, and 13 samples that only had 16S rRNA data (Fig 1C).

## There were large differences in the diversity and assemblages of the pathogenic and healthy disease inoculations used

The disease inoculation prokaryiomes clustered by disease inoculation type (WBTi DS, SM, HTS), with PC1 and PC2 explaining 75% of the variance (S2A Fig). Alpha diversity estimates identified more diverse prokaryiomes in the WBTi DS inoculations compared to the HTS (p<0.001) and *SM* (p<0.001) inoculations (S2B Fig). RA analysis revealed that *SM* inoculations were dominated by *Serratia* (S2C Fig), while the HTS was dominated by the genus MD3-55 (S2C Fig). The WBTi DS samples had bacteria found in other coral disease microbiome studies that were classed as putative pathogens: *Vibrio*, *Shimia*, *Oceanospirallum*, *Ferrimonas*, *Kordia*, and *Algicola*. However, there were differences between day-0 and day-3 in WBTi DS inoculations which included: *Agaribacter*, *Neptumonas*, and *Pseudoteredinibacter*. *Serratia* ASVs were also present in both day-0 (12%) and day-3 (11%) of the WBTi DS inoculations (S2C Fig).

## The short-term heat stress did not affect physiological metrics but did elicit a weak response from the coral host and prokaryiome

There were no significant differences (alpha<0.05) in I-PAM (S3B Fig) or buoyant weight (S3C Fig) measurements between ambient or STHS treatments. There were also no significant increases in risk for developing disease signs due to temperature treatment and disease inoculations within each genet (S4A Fig) or between the genets (S4C Fig). In contrast, disease inoculations independent of temperature had a significant difference in risk of disease signs developing between *Serratia* RR and WBTi RR for genet ML2 (S4B Fig).

For the coral hosts transcriptomic response, PCA did not show any clear separation in the 95% confidence intervals for all (S5A Fig), pre-exposure (S5B Fig) or disease inoculated (S5C Fig) coral samples. Differential expression analysis identified 28 significantly differentially expressed genes (DEG) for the pre-exposure coral samples (S3 File) with hierarchical clustering of these genes causing separation between ambient and STHS coral samples (S5D Fig). For disease inoculated samples, there were 21 significant DEG which all showed positive log 2-fold changes (L2FC) (S3 File) but no clear clustering between ambient and the STHS (S5E Fig).

PCA of the prokaryiome did not show any clear separation of 95% confidence intervals for all (S6A Fig), pre-exposure (S6B Fig), or disease inoculated (S6C Fig) corals. PERMANOVA analysis did however identify a significant difference between the compositions of ambient and STHS ($p < 0.001$) corals. Differential abundance analysis identified eight ASVs with significantly increased abundance, and nine ASVs with significantly decreased abundance in the pre-exposure STHS coral samples (S4 File). There was a significant difference in Shannon-

wiener estimates for the pre-exposure coral samples (p < 0.01), with STHS coral samples showing a more diverse prokaryiome than ambient corals (S6E Fig).

### The coral host shows similar transcriptomic profiles for visually healthy and diseased corals regardless of the disease inoculations used, with this characterized by increased immune activity, wound healing, and metabolic processes

When accounting for genet variance, PCA analysis identified coral samples grouping by the overall VHS (Fig 2A) rather than by the four different disease inoculations that were used (Fig 2B). PERMANOVA analysis corroborated this result with significant differences between pre-exposure and visually healthy (p-adj = 0.03, R2 = 0.07), pre-exposure and diseased (p-adj = 0.03, R2 = 0.11), and visually healthy and diseased (p-adj = 0.03, R2 = 0.17) coral samples, but no significant differences between the disease inoculation within the visually healthy and diseased corals. To explore the groupings due to VHS observed in PCA, a co-expression analysis using WGCNA was used. WGCNA identified 16 co-expression modules (S7 Fig) that ranged from 54 to 5,064 genes (S5 File). Of the 16 modules, six modules (Black, Dark Turquoise, Dark Green, Grey 60, Blue, and Light Cyan) showed significant correlations (alpha < 0.05) with pre-exposure, visually healthy, and diseased coral samples (Fig 2C). There was no difference in module correlation directionality when splitting the visually healthy corals by their respective disease inoculation (WBTi DS, HTS, *SM*, *SP*) compared to all visually healthy samples grouped together (Fig 2C). Similarly, diseased corals split by disease inoculation (WBTi DS & *SM*) also showed the same module correlation directionality when compared to all diseased samples grouped together (Fig 2C).

The Blue co-expression module (genes = 2,340, hub gene = *Ribosomal protein S6 kinase alpha-3*, Fig 2C & 2D) showed significant negative correlations with pre-exposure ($R^2$ = -0.21) and visually healthy ($R^2$ = -0.3) corals and a significant positive correlation with diseased corals ($R^2$ = 0.74). GO enrichment identified eight Cellular Component, 10 Molecular Function, and 29 Biological Process (S6 File). Within Biological Process, this included parent terms linked to *Signaling* (GO:0023052) and *Signaling Pathways* (GO:0023033), with child terms linked to *G-protein coupled receptor signaling pathways* (GO:0007186) and *Roundabout Signaling Pathway* (GO:0035385). Additionally, enrichment of terms involved in the *Response to Chemical Stimulus* (GO:0042221) and the *Response to Organic Cyclic Substance* (GO:0014070) were also present (S6 File). Cellular Component identified genes within the *Membrane* (GO:0016020) and *Plasma Membrane* (GO:0005886) to also be important. (S6 File). The Blue Module also showed the highest overlap (573 genes) with an innate immune module identified from previous transcriptomic disease research in *A. palmata* [45]. This included common genes such as the previously reported hub-gene (*D-amino acid oxidase*) as well as antimicrobial peptides (Achacin), c-type lectins, TNFs, toll like receptors and transcription factors (S6 File). From enrichment and comparison with the immune module from [45], the inferred function of the Blue module was deemed "Immune Processes". For a full list of overlap genes from the Blue module and from the Blue module and [45], please see S6 File.

The Black (genes = 1,961, hub gene = *SH3 domain-binding glutamic acid-rich-like protein 3*, Fig 2C & 2D) and Dark Green (genes = 657, hub gene = *Uncharacterized protein LOC110044255*, Fig 2C & 2D) co-expression modules showed positive correlations with visually healthy corals (Black $R^2$ = 0.34, Dark green $R^2$ = 0.79) and negative correlations with pre-exposure (Black $R^2$ = -0.2, Dark green $R^2$ = -0.5) and diseased (Black $R^2$ = -0.19, Dark green $R^2$ = -0.39) corals. The Black module showed significant GO enrichment for two Cellular Component, nine Biological Process, and 11 Molecular Function (S6 File). Two main functions of GO

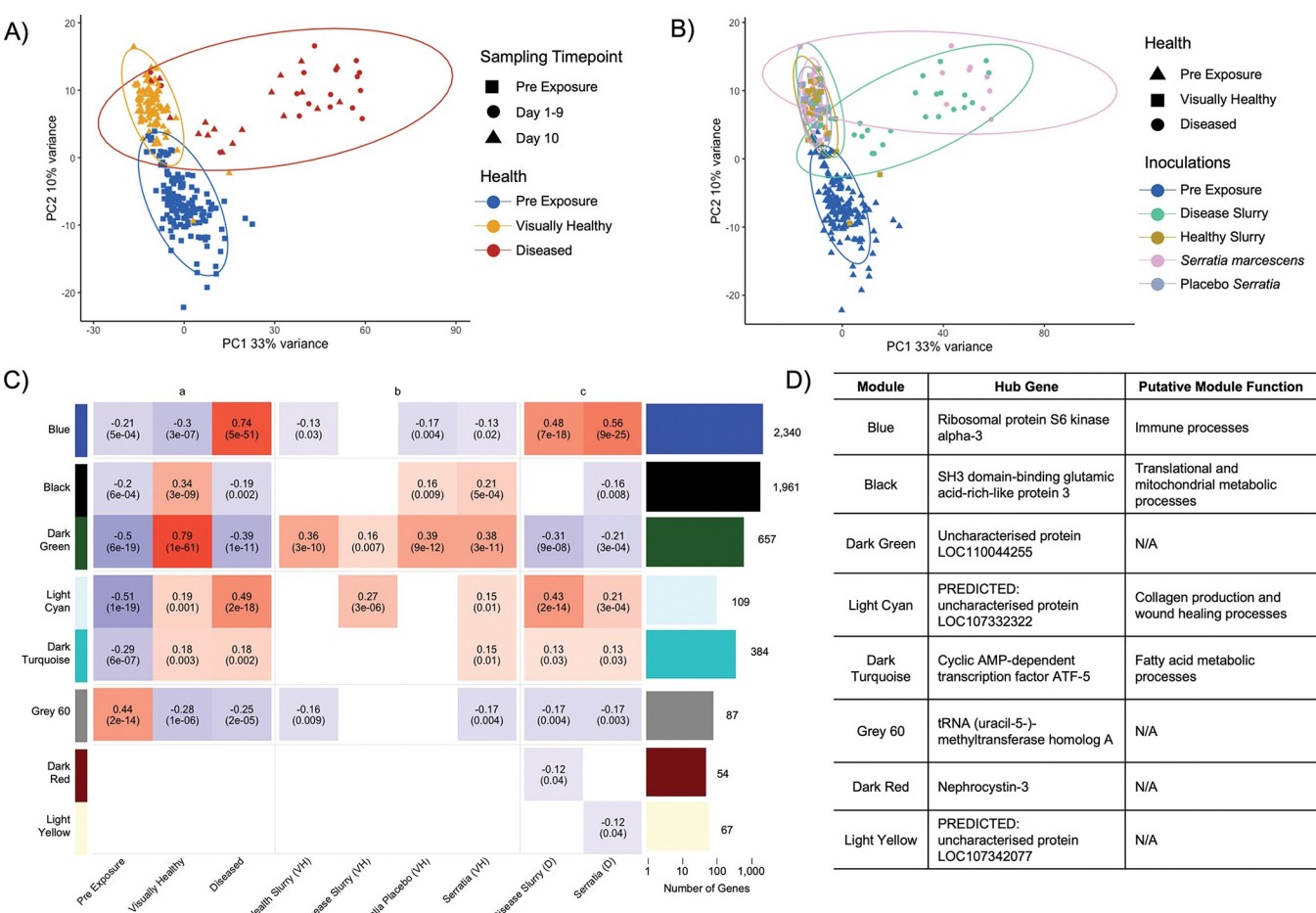

**Fig 2. There is a common response to disease inoculation at the transcriptomic level for the coral host.** A) Principal component [PC] 1 (33% variance) and PC2 (10% variance) are shown with samples split by visual health status [VHS] (blue = pre-exposure, yellow = visually healthy, dark red = diseased) sampling timepoint (square = pre-exposure samples taken before disease inoculations, circle = samples taken days 1–9, triangle = samples taken on day 10). B) PC1 (33% variance) and PC2 (10% variance) are shown with samples split by VHS (triangle = pre-exposure, square = visually healthy, circle = diseased) and disease inoculations (blue = pre-exposure, green = WBTi disease slurry [WBTi DS], mustard = healthy tissue slurry [HTS], pink = *Serratia marcescens* [*SM*], grey = *Serratia* placebo [*SP*]). C) Heatmap showing 8 of the 16 identified co-expression modules from WGCNA. Each co-expression module is a row with name color designation identified to the left. Columns are correlated metadata variables to co-expression modules and are split in three: a = coral samples grouped by VHS (control, visually healthy, diseased), b = visually healthy samples split by disease inoculations (WBTi DS, HTS, *SM*, *SP*), c = diseased coral samples split by disease inoculation (WBTi DS and *SM*). Within each heatmap cell, the top number shows the correlation of the trait with the identified co-expression module and the bottom number shows the significance of this correlation. Heatmap fill color is based on strength of correlation, with positive correlations being red, and negative correlations being blue. Cells that did not show significance at 0.05 were removed and replaced with empty white fill. Bar plot to the right of the heatmap shows the number of genes present in each co-expression module. Each bar color corresponds to co-expression module color and is aligned to respective co-expression module rows in the heatmap. Y-axis for the bar graph is on a log-10 scale. Numbers next to each bar identify the number of genes identified in each co-expression module. A-C used variance stabilized transformation [VST] counts with genet variance removed. D) Table showing 8 of the 16 co-expression modules with their hub gene and putative function identified from GO and KEGG enrichment analyses. For full lists of enriched GO and KEGG pathway terms please see S6 and S7 Files respectively.

terms were identified from enrichment: (1) genes linked to ribosomes and subsequent translational processes (S6 File), and (2) genes linked to mitochondria oxidoreductase and biosynthetic processes (S6 File). Like GO analysis, KEGG enrichment pathways associated with the *Ribosome* (KO:03010), were also identified. In addition, KEGG analysis also found *Thermogenesis* (KO:04714), and *Oxidative Phosphorylation* (KO:00190) (S7 File) to be significantly enriched. The inferred function for the Black module was "translational and mitochondrial metabolic processes". For the Dark Green module, GO enrichment analysis identified one

enriched term linked to Cellular Component, *Extracellular Regions* (GO:0005576) (S6 File), with no significant KEGG enrichment. Due to low enrichment results, no inferred function was assigned.

The Light Cyan (genes = 109, hub gene = *PREDICTED*: *uncharacterized protein LOC107332322*, Fig 2C & 2D) and Dark Turquoise (genes = 384, hub gene = *Cyclic AMP-dependent transcription factor ATF-5*, Fig 2C & 2D) co-expression modules showed significant negative correlations with pre-exposure corals (Light Cyan $R^2$ = -0.51, Dark Turquoise $R^2$ = -0.29) and significant positive correlations with visually healthy (Light Cyan $R^2$ = 0.19, Dark Turquoise $R^2$ = 0.18) and diseased (Light Cyan $R^2$ = 0.49, Dark Turquoise $R^2$ = 0.18) corals. The Light Cyan module showed no significant KEGG pathway enrichment. There was significant GO enrichment for two Biological Process terms, *Cell-Substrate Junction Assembly* (GO:0007044) and the child term *Hemidesmosome Assembly* (GO:0031581). There were also eight significant Cellular Component terms with these also linked to the *Hemidesmosome* (GO:0030056) and the *Basal Plasma Membrane* (GO:0009925), or the *Extracellular Region* (GO:0005576), specifically the child term *Collagen* (GO:0005581) (S6 File). From enrichment analysis, the Light Cyan module function was designated as "Collagen Production/Wound Healing". The Dark Turquoise module showed no significant GO enrichment. Significant KEGG pathway enrichment identified terms important in Fatty Acid Metabolic processes, as well as *Apoptosis* (KO:04210) and *Autophagy (Yeast)* (KO:04138) (S7 File). The inferred function for the Dark Turquoise module was designated as "Fatty Acid Metabolism".

## Disease inoculations showed increases in alpha diversity for the prokaryiomes of visually healthy and diseased corals

PCA analysis showed coral samples grouping by VHS (Fig 3A) with no unique profiles identified within the visually healthy or diseased corals due to disease inoculation (Fig 3B). PERMANOVA analysis corroborated this observation as significant differences were found between pre-exposure and visually healthy (p-adj = 0.03, $R^2$ = 0.104), pre-exposure and diseased (p-adj = 0.03, $R^2$ = 0.111) and visually healthy and diseased corals (p-adj = 0.03, $R^2$ = 0.079). There were, however, no significant differences within the visually healthy or diseased corals due to the different disease inoculations administered. Pre-exposure corals showed a prokaryiome with the lowest diversity (Shannon-Weiner = 3.77; Fig 3C) and were dominated by ASVs in the genera *Spirochaeta 2* (33.81%) and *Pseudomonas* (12.31%) (Fig 3D). The prokaryiomes of visually healthy corals were significantly more diverse (Shannon-Weiner = 4.53) when compared to pre-exposure corals (Shannon-Weiner = 3.78, $p < 0.01$) (Fig 3C). RA analysis identified similar patterns in pre-exposure corals, with the genera *Spirochaeta 2* (WBTi DS = 23.64%, HTS = 43.95%, *SM* = 32.82%, *SP* = 31.10%) and *Pseudomonas* (WBTi DS = 4.90%, HTS = 8.32%, *SM* = 16.22%, *SP* = 8.69%) showing the highest abundances (Fig 3D). The prokaryiomes of diseased corals were significantly more diverse (Shannon-Weiner = 4.24) compared to pre-exposure corals (Shannon-Weiner = 3.78, $p < 0.01$), but less diverse than visually healthy corals (Shannon-Weiner = 4.53, p < 0.01; Fig 3C). For the diseased corals, there was a higher relative abundance of ASVs in the genera *Vibrio* and (WBTi DS 8.87%, *SM* = 12.85%) and *[Caedibacter] taeniospiralis group* (WBTi DS = 14.73%, *SM* = 26.28%) compared to pre-exposure and visually healthy corals (Fig 3D). The genus *Microscilla* was only present in the WBTi DS samples (17.31%; Fig 3D). A full list of RA percentages are available in S8 File.

Co-abundance analysis identified six modules (S8 Fig), with three of these modules (Purple, Green, and Cyan) showing significant correlations to pre-exposure, visually healthy, and diseased corals (Fig 4A). The Purple co-abundance module (ASVs = 61, hub ASV = *Microscilla* spp, Fig 4A & 4B) showed significant positive correlations to diseased corals ($R^2$ = 0.55) and

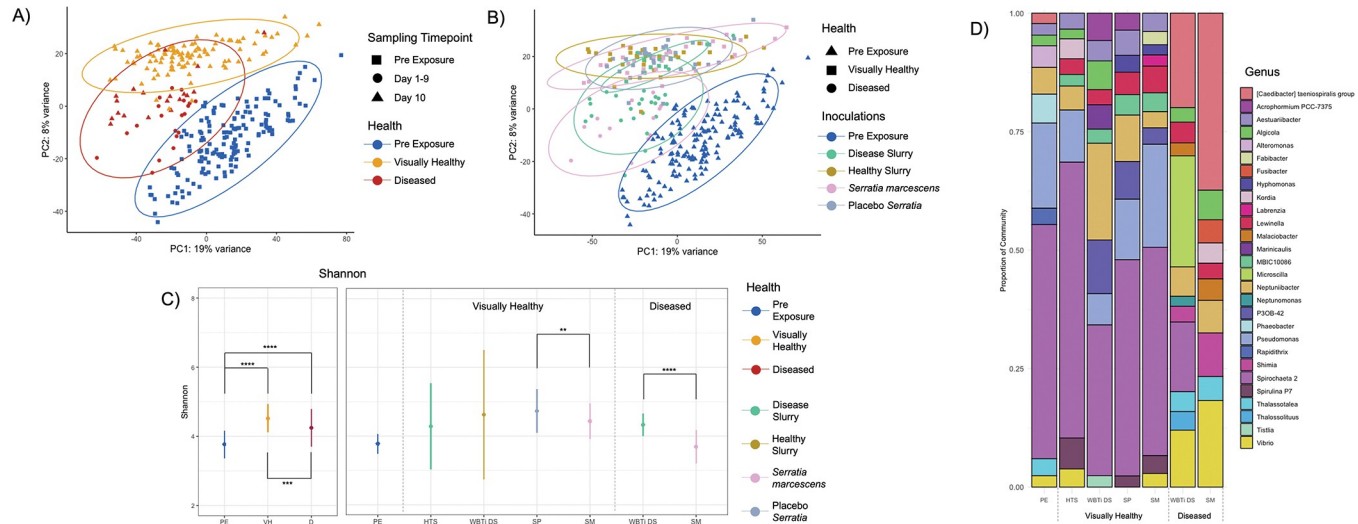

**Fig 3. The prokaryiomes has a similar shift and increased diversity in visually healthy and diseased corals, regardless of disease inoculation type.** A) Principal component [PC] 1 (19% variance) and PC2 (8% variance) with samples split by visual health status [VHS] (blue = pre-exposure, yellow = visually healthy, dark red = diseased) and sampling timepoint (square = pre-exposure samples taken before disease inoculations, circle = samples taken between day-1 and day-9, triangle = samples taken on day-10). B) PC1 (19% variance) and PC2 (8% variance) with samples split by VHS (triangle = pre-exposure, square = visually healthy, circle = diseased) and disease inoculation (blue = pre-exposure, green = WBTi disease slurry [WBTi DS], mustard = healthy tissue slurry [HTS], pink = *Serratia marcescens* [*SM*], grey = *Serratia* placebo [*SP*]). A-B are PCA analysis of the 16S rRNA center log ratio transformed (CLR) counts. C) Shannon-Weiner alpha metric (y-axis) grouped by VHS (x-axis). Circles show mean estimates for all samples in each respective VHS group, and error bars show confidence intervals. D) Relative abundance [RA] for samples within each VHS (x-axis), and proportions of each genus identified from RA analysis (y-axis). Bar fills = different genus (annotated using the SILVA database) showing the largest contributions to the RA, with color to genus fill identified by legend. For C- D: PE = pre-exposure, VH = visually healthy, D = diseased, HTS = healthy tissue slurry, WBTi DS = white band type I disease slurry, SP = *Serratia* placebo, SM = *Serratia marcescens*. For C, stars signify significance levels (* = <0.05, ** = <0.01, *** = <0.001, **** = < 0.0001) calculated using Betta.

significant negative correlations to both pre-exposure ($R^2$ = -0.21) and visually healthy ($R^2$ = -0.17) corals. Of the 61 ASVs, 25 ASVs annotated to *Microscilla* spp, seven ASVs to *[Caedibacter] taeniospiralis group* spp, and three ASVs to *Thalossolituus* spp (S9 File). There were also ASVs annotated to a *Malaciobacter* spp, *Candidatus Megaira* spp, *Peredibacter* spp, and *Pseudobacteriovorax* spp (S9 File).

The Green co-abundance module (ASVs = 103, hub ASV = *Spirochaeta 2* spp, Fig 4A & 4B) showed significant positive correlations to visually healthy ($R^2$ = 0.33) and diseased corals (R2 = 0.33) and a significant negative correlation to pre-exposure corals ($R^2$ = -0.54). Of the 103 ASVs, 26 annotated to *Spirochaeta 2* spp, and three to *Aestuariibacter* spp (S9 File). Other ASVs annotated to a *MDS-55* spp, *Corynebacterium* spp, a *Terasakiella* spp, a *Planctomicrobioum* spp, and a *Peredibacter* spp (S9 File).

The Cyan co-abundance module (ASVs = 2,978, hub ASV = PUTATIVE Phylum Bdellovibrionata, Fig 4A & 4B) showed a positive correlation with pre-exposure ($R^2$ = 0.31) and increasing negative correlations with visually healthy ($R^2$ = -0.1) and diseased corals ($R^2$ = -0.31) corals. The phylum Proteobacteria dominated annotated ASVs (n = 832, 44.35%) with 305 annotating to Alphaproteobacteria and 512 annotating to Gammaproteobacteria (S9 File). For the alphaproteobacterial, the most dominant family was Rhodobacteraceae (25.83%) (S9 File). The order Rhizobiales accounted for 13.25% of ASVs with this split between the families Beijerinckiaceae, Hyphomicrobiaceae, Devosiacea, Methylogellaceae, and unannotated ASVs (S9 File). The order Caulobacterales accounted for 12.91% of ASVs split between three families: Hypomonadaceae, Caulobacteraceae, and Parvularculaceae (S9 File). ASVs within the orders Sphingomonadales (5.70%), Rhodospirllales (4.64%) and Rickettsiales (4.30%) were also present in high abundance (S9 File). The Gammaproteobacteria had 39.10% of ASVs not

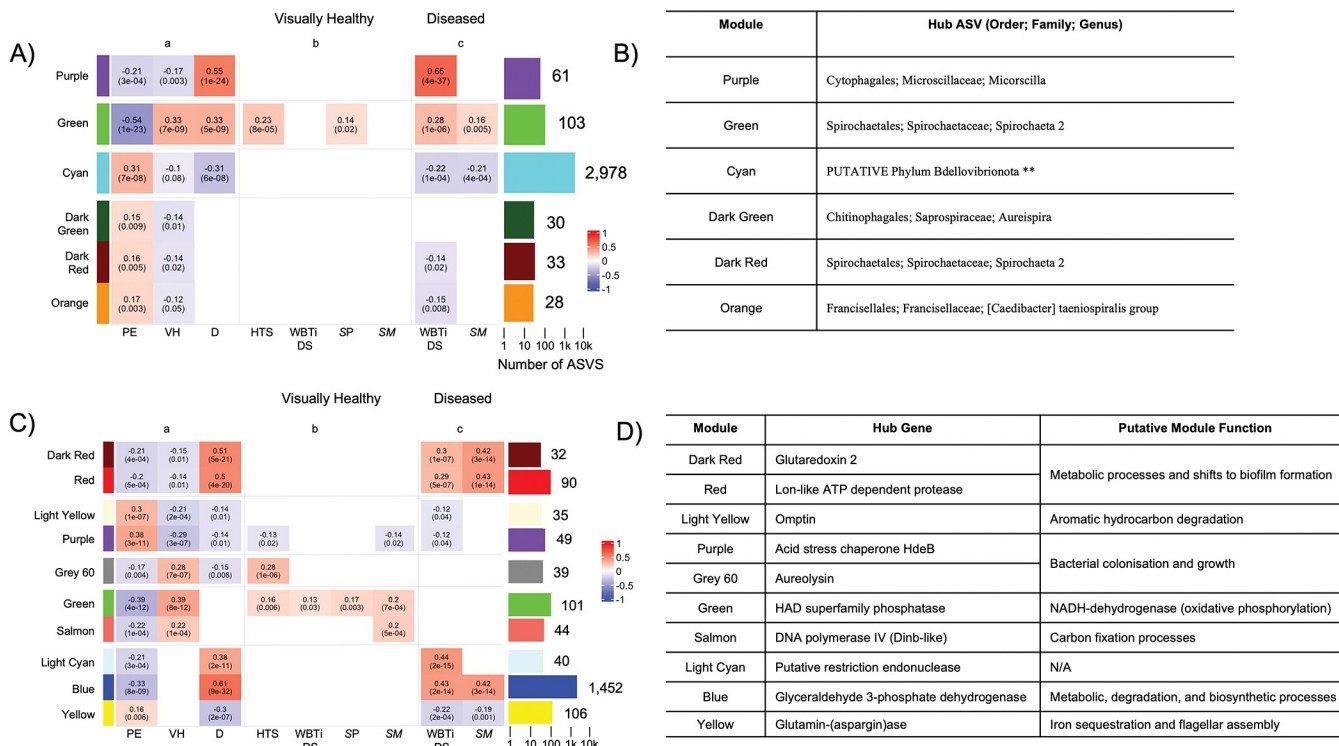

**Fig 4. Significantly correlated prokaryotes to pre-exposure, visually healthy and diseased coral samples, with inferred metagenomic function identifying potential increased metabolic processes, and shifts to biofilm states, in disease inoculated coral samples.** A) Heatmap showing the six identified co-abundance modules identified from WGCNA analysis using the center log ratio transformed counts. Each co-abundance module is a row, with module name and color designation identified to the left of the heatmap. B) Table showing the 6 co-abundance modules with their identified hub ASV. C) Heatmap showing ten of the 18 co-expression modules of inferred metagenomic function of the 16S rRNA data using PICRUSt2. Each identified co-expression module is a row, with module name and color designation identified to the left of the heatmap. D) Table showing the 10 of 18 inferred metagenomic function [IMF] co-expression modules with their hub gene and putative function identified from KEGG enrichment analyses. For A) and C): columns are metadata traits that were correlated with identified co-abundance and IMF co-expression modules: a = grouped visual health status [VHS] (control, visually healthy, diseased), b = visually healthy samples split by disease inoculation (WBTi DS, HS, *SM*, *SP*), c = diseased samples split by disease inoculation (WBTi DS, *SM*). Bar graph to the right shows the number of ASVs (A) and inferred genes (C) within each module, with color matching module color and position in the heatmap. X-axis is on a log scale, with the number of ASVs (A) and genes (C) identified at the end of each bar. Heatmap cell fill shows positive correlations (red) to negative correlations (blue) with correlation between modules and metadata traits. For each cell, the first number shows the strength of the correlation, the lower value (in parenthesis) shows significance of the correlation.

annotating to a higher taxonomic class (S9 File). For the gammaproteobacteria, the order Oceanospirillales (13.66%) showed the highest proportion of ASVs, with these ASVs in the families Alcanivoracaceae, Endozoicomonadaceae, Halamonadaceae, Kangiellaceae, Nitrinco-laceae, Oceanospirillaceae, Oleiphilaceae, Pseudohongiellaceae, and Saccharospirillaceaea (S9 File). There were also ASVs in the orders Cellvibrionales (10.30%), Alteromonadales (9.03%), Pseudomonadales (4.41%), Vibrionales (2.94%), and Enterobacterales (2.73%). Four ASVs also annotated to the genus *Serratia* (S9 File).

## Co-expression analysis of the inferred metagenomic function of the prokaryiome identifies increased metabolic processes and shifts to biofilm states in disease inoculated corals

WGCNA analysis of the IMF identified 18 co-expression modules (S9 Fig). There were two modules, the Dark Red module (genes = 32, hub gene = *Glutaredoxin 2*, Fig 4C & 4D) and Red module (genes = 90, hub gene = *Lon-like ATP dependent protease*, Fig 4C & 4D), which were significantly correlated with diseased corals (Dark Red $R^2$ = 0.51, Red $R^2$ = 0.5), and negatively

correlated with pre-exposure (Dark Red $R^2$ = -0.21, Red $R^2$ = -0.2) and visually healthy (Dark Red $R^2$ = -0.15, Red $R^2$ = -0.14) corals. The Dark Red module showed significant KEGG pathway enrichment of the *Two component system* (KO:02020), *Biofilm formation–Pseudomonas aeruginosa* (KO:02025) and *Starch and sucrose metabolism* (KO:00500) (S10 File). Similarly, the Red IMF module showed significant KEGG pathway enrichment of the *Two component system* (KO:02020) and *Biofilm formation–Escherichia coli* (KO:02026), with the other eight terms important in metabolic and fixation processes (S10 File). From enrichment results, the putative function of the Dark Red and Red modules was inferred as "Metabolic processes and shifts to biofilm formation".

The Light Yellow (genes = 35, hub gene = *Omptin*, Fig 4C & 4D) and Purple (genes = 49, hub gene = *Acid Stress chaperone HdeB*, Fig 4C & 4D) IMF co-expression modules showed significant positive correlations with pre-exposure corals (Light Yellow $R^2$ = 0.3, Purple $R^2$ = 0.38) and significant negative correlations with visually healthy (Light Yellow $R^2$ = -0.21, Purple $R^2$ = -0.14) and diseased (Light Yellow $R^2$ = -0.14, Purple $R^2$ = -0.14) corals. There was significant enrichment of eight KEGG pathway terms for the Light-Yellow IMF module, with all terms important in aromatic hydrocarbon degradation processes (S10 File). From enrichment, the purple module was assigned the putative function "Aromatic hydrocarbon degradation." The Purple IMF module showed significant KEGG pathway enrichment of three terms: Propanoate metabolism (KO:00640), Selencompound metabolism (KO:00450), and Biofilm formation–Escherichia coli (KO:02026) (S10 File). Enrichment analysis led to the Purple module's putative function being linked to "Bacterial colonization and growth".

## The coral hosts immune, wound healing, and fatty acid metabolic processes are significantly correlated to disease state prokaryiomes which have increased growth and metabolic function

The Blue "Immune Processes" transcriptomic co-expression module showed positive significant correlations with the Purple ($R^2$ = 0.42) and Green ($R^2$ = 0.26) co-abundance modules which showed positive correlations with diseased corals. (Fig 5). There were also significant positive correlations to the Dark Red (R2 = 0.52), Red ($R^2$ = 0.7) and Blue ($R^2$ = 0.48) IMF co-expression modules (Fig 5) which were enriched for prokaryotic growth and metabolic processes.

The Light Cyan "Collagen Production and Wound Healing" transcriptomic co-expression module also showed significant positive correlations to the Purple ($R^2$ = 0.31) and Green ($R^2$ = 0.32) prokaryotic co-abundance modules (Fig 5). There were also significant correlations with the Dark Red ($R^2$ = 0.25), Green ($R^2$ = 0.27), and Blue ($R^2$ = 0.31) IMF co-expression modules (Fig 5) with these enriched in prokaryotic growth and metabolic functions.

The Dark Turquoise "Fatty Acid Metabolism" transcriptomic co-expression module only showed a significant positive correlation with the green ($R^2$ = 0.21) prokaryotic co-abundance module (Fig 5). There were also significant positive correlations with the Dark Red ($R^2$ = 0.18), Red ($R^2$ = 0.2), Green ($R^2$ = 0.17), and Blue ($R^2$ = 0.23) IMF co-expression modules (Fig 5), with these IMF modules again enriched for prokaryotic growth, metabolism, and NADH dehydrogenase genes.

## Differences in baseline coral host immune gene expression drove the differences between genets, while there was a core prokaryiome between all genets

PCA identified that genet explained the largest amount of variance for the transcriptomic data, with significant correlations to PC2 (13%), PC3 (8%), and PC5 (6%) (S10A Fig). Visualization of PC2 and PC3 showed clear clusters of genet identity (S10B Fig), and 6,225 significantly

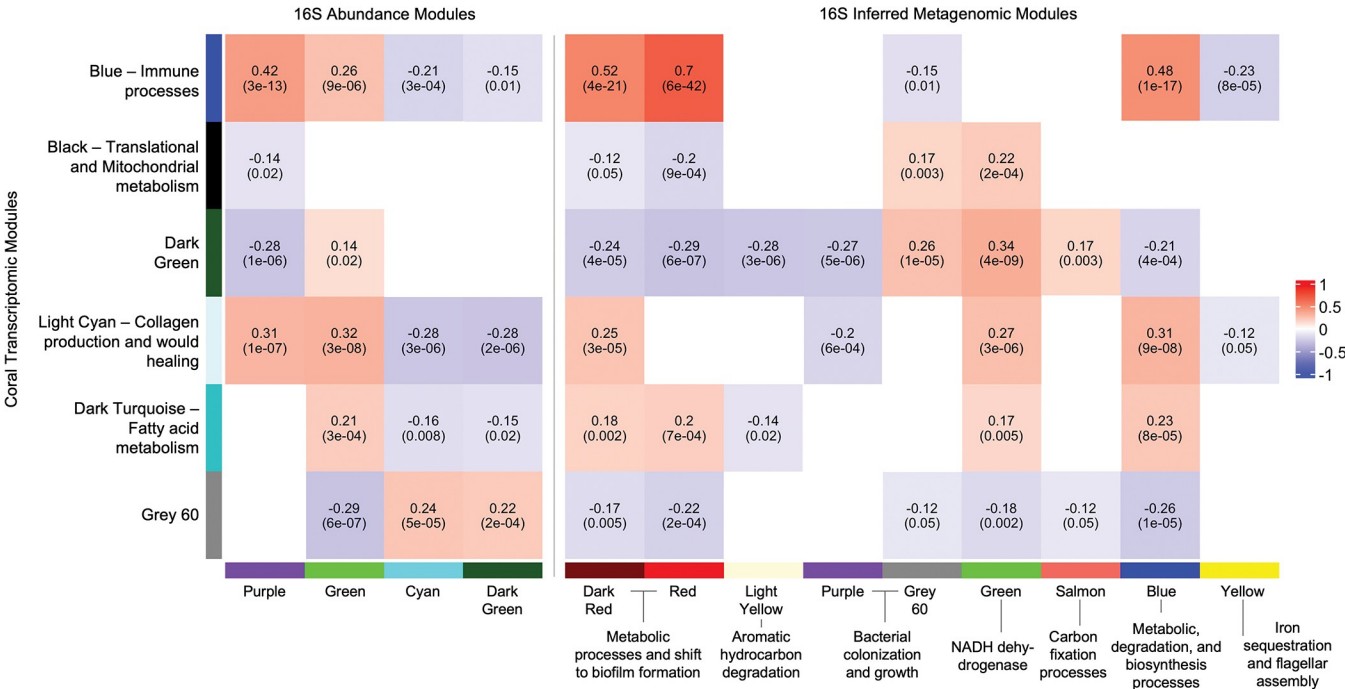

**Fig 5. There are positive correlations between the corals hosts immune, wound healing, and metabolic processes with the prokaryiomes disease associated ASVs and shifts to biofilm states.** Heatmap correlating the putative host transcriptomic function co-expression modules (rows) to either 16S rRNA co-abundance (column left), or putative functions from the inferred metagenomic function [IMF] co-expression modules (column right). Heatmap cell fill shows positive correlations (red) to negative correlations (blue) between calculated eigengenes for the transcriptomic and 16S rRNA datasets. For each cell, the top number shows the strength of the correlation, and the bottom number shows the significance of the correlation.

differentially expressed genes were identified (S11 File). GO enrichment analysis identified 10 significantly enriched terms, all of which were in the GO category *Biological Process* (S11 File). Three terms were linked to immunity: *Immune Response* (GO:0006955), *Innate Immune Response* (GO:0045087), and *Regulation of Immune Response* (GO:0050776). Visualization of the top 75 most significant genes within these immune GO terms identified strong hierarchical clustering of samples by genet identity and gene expression profiles (S10C Fig). Within the top 75 most significant genes there were a range of immune receptors: NOD-like receptors, C-type lectins, and tumor necrosis factors (TNFs) (S10C Fig). There were also genes involved with immune signaling pathways: elf-2 kinases, Toll pathway associated genes, and aminopeptidases (S10C Fig). The remaining seven enriched GO terms were all associated with metabolic and biosynthetic processes: mycothiol biosynthetic (GO:0010125) and mycothiol metabolic (GO:0010126) processes, glycoside biosynthetic (GO:0016138) and glycoside metabolic (GO:0016137) processes, oligosaccharide metabolic processes (GO:0009311) and disaccharide metabolic processes (GO:0005984; S11 File).

For the coral prokaryiome, genet identity did not strongly correlate to any PCs explaining large amounts of variance (PC4 = 3% and PC5 = 2%) (S11A & S11B Fig). RA analysis did however identify differences in pre-exposure coral samples when split by genet (S11C Fig). *Spirochaeta 2* spp showed high abundance in genets CN2 (24%), HS1 (39%) and ML2 (72%) and low abundance in CN4 (1%) (S11C Fig). *Pseudomonas* spp were also present in all genets, but again at differing abundances (CN2 = 15%, CN4 = 22%, HS1 = 8%, ML2 = 3%). Analysis of the core microbiome of the pre-exposure coral samples identified 51 ASVs (S12 File) with four ASVs showing the highest prevalence: *Spirochaeta 2* spp (ASV_4), Family Oxalobacteraceae (ASV_13), *Pseudomonas* spp (ASV_17), and a *Phaeobacter* spp (ASV_11).

## Discussion

### Different disease inoculations cause common expression profiles for visually healthy and diseased corals

The transcriptomic immune response to diseases has been well characterized in other coral species (reviewed in [36]) as well as in A. palmata [45], but how a coral species reacts to different disease inoculations has not yet been identified in an experimental setting. The coral hosts' transcriptomic response showed the same expression profiles for visually healthy or diseased corals regardless of which disease inoculation was administered as identified through PCA (Fig 2A) and co-expression (Fig 2C) analysis. Specifically, there was a core immune response that was initiated, identified through co-expression analysis (Blue module), regardless of disease inoculation type, with this including a wide range of innate immune signaling pathways (S6 & S7 Files). Our results here indicate that once a disease is initiated, the coral may have a general immune response despite differing pathogenic sources. A previous meta-analysis identified a similar pattern, where the coral host initiates a core immune response to a wide range of stressor types which included disease inoculation [106]. Additional evidence for this argument comes from the Blue co-expression module which also showed the highest overlap with the immune module from a previous transcriptome co-expression disease study in *A. palmata* [45]. This included D-amino acid oxidase, antimicrobial peptides, TNF receptors, c-type lectins, and NOD-like receptors all of which are important in the innate immune system. Consistent identification of these genes over time and space indicates a core response to potentially different pathogenic material in *A. palmata* that may allow generation of health biomarkers in this coral species.

Co-expression analysis also identified increased collagen and wound healing processes in visually healthy and diseased corals (Fig 2C). Disease inoculated corals may also increase collagen production and deposition to arrest open wounds at the disease lesion. Collagen provides support to the extracellular space for connective tissues, making it a key component of healthy tissue in organisms [107] as well as important in the wound healing process [108]. Increased collagen deposition is found in the model wound healing invertebrate *Hirudo verana* [109], as well as in the Cnidarian *Calliactis polypus* [110]. Also enriched in disease inoculated corals were the GO terms *hemidesmosome* (GO:0030056) and *hemidesmosome assembly* (GO:0031581), which facilitate the stable adhesion of cells to the underlying membrane [111] and are also integral to wound healing processes in vertebrates [112,113].

Increased fatty acid metabolism may also be playing a key role in initiating and maintaining the immune response in *A. palmata*, with increased expression identified in visually healthy and diseased corals (Fig 2C). Fatty acids comprise a key form of energy storage in a wide range of animals [114] and, when fully metabolized, provide the highest yield of energy compared to other macronutrients [115]. Therefore, this could indicate that disease inoculated *A. palmata* increase fatty acid metabolic processes to provide additional energy resources to energetically expensive immune responses to fight disease [41,45]. Fatty acid metabolism could be an indicator of disease resistance and susceptibility in corals, with individuals/genets having higher lipid and fatty acid reserves able to mount stronger and longer immune responses. While not yet tested in relation to disease, increased lipid diets do reduce the impact of thermal stress in corals [116] and could also therefore influence the risk of manifesting disease signs.

### The prokaryiome of disease inoculated *A. palmata* shows common disease states indicating opportunistic prokaryotes may be important in the final stages of disease infection

Like the coral host's transcriptomic response, the prokaryiome also showed similar communities for visually healthy and diseased corals, through PCA (Fig 3A) and co-abundance (Fig 4A)

analyses despite the different disease inoculation administered. This indicates that we may have captured successional states of the prokaryiome after initial infection from pathogenic sources. The successional state hypothesis of coral disease is highlighted in [16], where the authors hypothesize that primary pathogen(s) cause an initial destabilization, allowing unchecked growth of normally beneficial microbes, as well as colonization of foreign microbes. We also identified increasing alpha diversity in visually healthy and diseased corals (Fig 3C). Secondary infection, and visual signs of disease in corals, are commonly accompanied by increased alpha diversity metrics [51,52,117]. This indicates that we may have captured these opportunistic prokaryiome states in visually healthy corals, and the final disease state in diseased corals. This indicates that the prokaryiome could be a key indicator of coral health before visual signs of disease manifest.

Co-abundance analysis provided additional information indicating we may have captured different successional states of the prokaryiome in *A. palmata*. The Purple module was significantly correlated to diseased corals (Fig 4A) with the hub ASV identified as a *Microscilla* spp (Fig 4C). The *Microscilla* genus is within the order Cytophagales, a non-spore-forming, rod-shaped, gram-negative bacteria with a genome enriched with genes for nitrogen metabolism and the synthesis and utilization of phosphorous compounds [118]. Previous work in visually healthy corals showed an increase in relative abundance of Cytophagales after inoculation with *Vibrio coralliilyticus* [119]. *V. coralliilyticus* was the primary pathogen, which provided a disruption to the prokaryotic community, and then allowed opportunistic bacteria like Cytophagales to colonize the coral [119]. In our study, WBTi DS pathogen(s) may have also destabilized the coral prokaryotic community and resulted in the proliferation of *Microscilla* spp. High abundances of *Microscilla* spp. could potentially be used as a biomarker of WBTi disease in *A. palmata* since these ASVs were not significantly correlated to the *S. marscecens* inoculation (Fig 4A). The genus *[Caedibacter] taenospirales group* was also enriched in the diseased corals and present in very low abundances in pre-exposure and visually healthy corals (Fig 3D). Although the exact role of *Caedibacter* species has not been identified, its low abundance in pre-exposure corals could be another example of a shift from low abundance bacteria to opportunistic growth in corals exhibiting active disease signs.

Disease inoculated corals showed the same positive correlation with the Green co-abundance module (Fig 4A). The hub ASV for the Green co-abundance module was a *Spirochaeta 2* spp, an identified core microbiome member in pre-exposure coral samples (S12 File). Comparison of the *Spirochaeta 2* ASVs present in the Green co-abundance module to co-abundance modules that were significantly correlated with pre-exposure corals (Dark Green, Dark Red, Cyan, and Orange) showed no overlap. This, we hypothesize, could indicate specific *Spirochaeta 2* species are present in the *A. palmata* microbiome, and that different opportunistic species are present in the pathobiome state. Two ASVs annotated to *Corynebacterium* spp. were correlated to the disease inoculated samples (S9 File) with these identified as important in the human skin microbiome [120], reducing pathogen abundance and virulence [121,122], and activating the host immune response [123,124]. Despite this, *Corynebacterium* spp. have also been shown to act as opportunistic pathogens in mammals [125–127]. With visual signs of coral disease hypothesized to be the microbiome in dysbiosis with overgrowth of opportunistic microbes [16], *Corynebacterium* spp. could potentially act as a biomarker for corals that have been infected by a causative pathogen, with higher abundance indicative of the holobiont in a dysbiotic state. Future work should identify the function of *Corynebacterium* spp. in disease and identify whether it shifts from low abundance in the beneficial microbiome and becomes an opportunistic pathogen during disease.

## Co-abundance analysis identifies sets of healthy prokaryiome members in *A. palmata*

The Cyan, Dark Green, Dark Red and Orange co-abundance modules were only significantly correlated to pre-exposure corals (Fig 4A). This provides potential sets of prokaryotic members which are associated with the healthy *A. palmata* prokaryiome. The Dark Green module hub ASV was identified as an *Ausperia* spp (S9 File). *Ausperia* spp have previously been shown to potentially prey upon *Vibrio* spp [128] indicating that these bacteria may play a key role in regulating and maintaining low abundances of *Vibrio* that are commonly identified in healthy coral prokaryiomes [18,129,130]. There was also one ASV which annotated to a *Halobacteriovorax* spp in the Dark Green module (S9 File). *Halobacteriovorax* spp have been previously identified as a core microbiome member in another Caribbean coral, *Montastraea cavernosa*, functioning as a top-down control for invasive *Vibrio coralliilyticus* [119].

The Dark Red module hub ASV was identified as a *Spirochaeta 2* spp. This genus has been previously identified in the healthy prokaryiome of *Acropora palmata* [51] and may be functioning in nitrogen and carbon fixation [131]. With a *Spirochaeta 2* ASV also identified as a highly abundant member of the core microbiome in pre-exposure corals (S12 File), this indicates *Spirochaeta* 2 are most probably keystone members in the *A. palmata* prokaryiome. Across all the co-abundance modules which showed significant correlations with pre-exposure corals (Fig 4A), there were several ASVs annotated to the orders Flavobacteriaceae, Oceanospirillales, and Altermondales (S9 File) which have been identified as putative beneficial microbes in coral microbiomes [132]. Flavobacteriaceae have been shown to function as energy scavengers from organic debris [133], while Oceanospirillales species have been shown to be important in carbon fixation and sulfur oxidation [134–136]. Future work should look to expand on these sets of putative beneficial prokaryotic members and the role they play in the *A. palmata* microbiome.

## The prokaryiomes of disease inoculated corals show increased metabolic processes and shifts from free-living to biofilm states

IMF co-expression modules identified increased expression of genes linked to biofilm formation compared to a free-living state in the prokaryiome of visually healthy and diseased corals (Fig 4C), which has previously been shown to occur by abiotic or biotic stimuli [137]. Specifically, biofilm states are considered an essential factor in the pathogenesis of opportunistic bacteria [138] with shifts to biofilm states being more resistant to phagocytic processes [139,140] and antimicrobial agents [141,142] compared to equivalent planktonic states [140,143]. Biofilms have been characterized in other coral disease work [144,145] and are hypothesized to be a later successional stage after infection and destabilization of the coral by a primary pathogen [16]. Our results indicate a similar pattern may be occurring in *A. palmata*, with shifts from free living to biofilm formation occurring in disease inoculated corals indicating a later successional stage of disease infection.

Visually healthy and diseased corals showed increased metabolic and degradation processes for identified IMF co-expression modules (Fig 4C). Increased metabolic processes are commonly associated with opportunistic prokaryotes taking advantage of favorable environmental conditions [146] causing a shift to a dysbiotic state. This dysbiosis can therefore open favorable environments, such as the coral mucus, for opportunistic prokaryotes to colonize and grow. Additionally, increased prokaryotic metabolic processes are also associated with switches to biofilm formation. Biofilm formation is an energetically expensive process and is commonly accompanied with increased metabolic rates of prokaryotic communities [147,148]. This provides additional evidence that we may have captured the later stage of disease response of *A*.

*palmata* which is characterized by colonization of opportunistic bacteria that are shifting to a biofilm state with increased metabolic processes.

## Increased host immune, wound healing, and fatty acid metabolism processes are significantly linked to pathogenic prokaryotic co-abundance modules and IMF co-expression modules enriched for metabolic processes and shifts to biofilm formation

Correlative analysis between the different omics WGCNA eigenvectors was used to provide an integrated analysis between the coral hosts gene expression and prokaryiome (Fig 5). Coral disease research has shown that the host immune system is activated by, and maintained, on disease challenge [37–47] with increases in prokaryotic diversity and potentially pathogenic members also occurring [51,52,117]. Despite this, cross-omics analyses have not yet been undertaken for the same piece of coral tissue in relation to disease work. Here, visually healthy and diseased corals were characterized by increased immune, wound healing, and fatty acid metabolic processes in co-expression analysis (Fig 2C) and our correlative analysis has provided disease associated co-abundance modules (Purple, Green) that house prokaryotic members that could be important in stimulating the corals immune response. Additionally, the IMF co-expression modules identified specific prokaryotic processes, namely metabolic and biofilm formation, which could result in additional initiation and maintenance of the coral immune system.

## Short term heat stress does not cause an increase in disease susceptibility in *Acropora palmata*

Previous studies have found increased disease susceptibility due to temperature stress and bleaching in coral species [28,29,149]. We wanted to test how STHS would influence disease susceptibility and risk without inducing visible bleaching signs. Through RR analysis, our results indicate that STHS does not increase disease susceptibility, or disease risk, in *A. palmata* in relation to the pathogenic inoculations used. Previous coral research has identified similar trends, where STHS events did not cause increases in coral disease incidence [150]. This indicates that longer term heat stress events may be required to increase disease susceptibility in *A. palmata*. This result could be of significance to coral restoration efforts in the Caribbean, as previous work has noted that *A. palmata* is more bleaching resistant than *A. cervicornis* [65,66,103] and thus may indicate it is a better species to use at warmer restoration sites. It is important to note however that the STHS used here may have not been long enough, or strong enough, to elicit the hypothesized increase in disease susceptibility. In-situ, *A. palmata* has been shown to increase bleaching incidence with cumulative heat stress at 31˚C for greater than 10-days [78]. This indicates that the strength and duration of the STHS used here was not enough to cause the early transcriptomic changes associated with the bleaching response, and thus caused no changes in disease susceptibility. Additionally, previous work in *A. palmata* identified clear transcriptomic profiles at a two-day STHS of 32˚C [151]. While the STHS used in this study was longer than the previous work in *A. palmata* [151], 30˚C may have been too low to initiate and cause the previously identified transcriptomic profiles.

Despite no significant transcriptomic profiles from the STHS, there was a significant difference in the prokaryiome between pre-exposure STHS and ambient corals with this subsequently lost in disease inoculated samples. Heat stress has been shown to cause large shifts in the coral microbiome [152–154], and our results here indicate that even STHS can start to influence the corals prokaryiome. This may provide a finer-scale diagnostic tool of coral health in relation to shorter and weaker heat stress events compared to using the coral hosts

transcriptomic response. Future work should look to incorporate a wider range of temperatures, as well as exposure times, to fully identify the effects of STHS on disease susceptibility.

## Baseline innate immune gene expression could drive patterns of disease resistance and disease susceptibility in *Acropora palmata*

Genet identity was the largest driver of the transcriptomic variance for the four genets of *A. palmata* used in this study (S10A & S10B Fig). Prior work has identified that genet identity can play an important role in patterns of disease resistance and susceptibility in coral species [28,29,43]. Here, we characterized immune genes that drove the variance between the four genets of *A. palmata* used (S10C Fig) and we therefore hypothesize that differences in baseline immune gene expression between genets of the same species could play an important role in disease resistance and susceptibility. For example, we identified differing baseline expression levels of NOD-like receptors and their signaling pathways between the four genets. NLRs and their signaling pathways are important in initiating the response to damage-associated molecular patterns (DAMPs) and pathogen-associated molecular patterns (PAMPs) [155,156] indicating that higher baseline expression of these pathways could confer increased disease resistance due to a primed and active immune system. It is important to note that while we did observe differences in baseline immune expression between the four genets, there were no significant differences in disease susceptibility identified. We therefore recommended that this observation be built upon utilizing a larger set of genets of *A. palmata* in a disease exposure study so to identify if 1) differences of baseline immune expression hold true in a larger sample size, and 2) if the differences of baseline immune expression do correlate and explain differences in disease susceptibility and resistance.

## A conserved core prokaryotic community was found in *A. palmata* over space and time

Coral species often show highly similar prokaryiomes that are also conserved over space and time [48,157]. This pattern held true in our study with the four genets of *A. palmata* showing a common and core prokaryiome. Core microbiome analyses identified an ASV from the family Oxalobacteraceae (S12 File) to have the highest detection threshold between the four genets. While the exact role of Oxalobacteraceae has not yet been identified in *A. palmata*, it is commonly found in the healthy mucus of other coral species [158,159] as well as being a core prokaryiome member in *A. cervicornis* [160]. Genome analysis of a marine Oxalobacteraceae isolate has shown that they possess one-carbon ($C_1$) metabolic genes [161] which indicates they may play a key role in metabolizing coral produced $C_1$ compounds.

Core microbiome analyses also identified a *Spirochaeta 2* ASV to be highly abundant between the four genets, with this corroborated in relative abundance analysis (S11C Fig). *Spirochaeta 2* have previously been identified in high abundance in healthy *A. palmata* prokaryiomes [51] indicating that *Spirochaeta 2* bacteria may be keystone members in this coral species. While the exact role in *A. palmata* has not yet been identified, *Spirochaeta 2* bacteria have been shown to be important in nitrogen and carbon fixation [131] and correlated with coral survivorship [162].

An ASV annotated to the genus *Pseudomonas* also showed high prevalence in the core microbiome analysis with this again evident in the relative abundance analysis (S11C Fig). Bacteria within the genus *Pseudomonas* are capable of metabolizing dimethyl sulfate compounds (DSCs) produced in large quantities by the Symbiodiniaceae [163,164], and they have been hypothesized to be beneficial microorganisms important in coral health [53]. Previous work in *A. palmata* however did not identify high proportions of *Pseudomonas* bacteria in the healthy

prokaryiomes, but instead, high proportions of bacteria within the genus *Endozoicomonas* (Family Oceanospirales) [51]. Bacteria within the Oceanospirales family have previously been shown to function in the coral carbon and sulfur cycle by metabolizing DSCs [165]. This, we hypothesize, indicates that DSC metabolism is a key trait of the coral prokaryiome, and in *A. palmata* the bacteria performing this role can be flexible. Future work should look to characterize the exact functional role of *Pseudomonas*, while also identifying whether prokaryiome members performing DSC metabolism can be fluid or are specifically selected for.

## Conclusions and future directions for disease work in *A. palmata*

Our work has identified that *A. palmata* reacts to different pathogenic sources with a common response which is characterized by immune, wound healing, and fatty acid metabolic processes. Additionally, the coral prokaryiome shifts to a more diverse state with an increased abundance of putative opportunistic microbes. A prudent next step would be to characterize whether this holds true, for the coral host and prokaryiome, with other pathogenic sources. The pathogenic (WBTi DS and *SM*) and HTS inoculations all housed gram-negative bacteria which may be key in the coral host and prokaryiomes response. Utilizing other pathogenic sources, such as viral or fungal, would allow identification of whether the hosts expression profiles, and prokaryotic communities, always shift to similar visually healthy or diseased states, or if there are specialized responses to different types of pathogenic organisms. Inclusion of control corals without any homogenate inoculation would also be an important consideration for future work, as this would allow identification of shifts in the coral hosts transcriptomic response, and prokaryiomes response caused by control inoculations (HTS and *SP*). Additionally, characterization of the entire coral microbiome's response to different disease inoculations should be incorporated. While we identified similar prokaryiome shifts in visually healthy and diseased corals, it is unknown whether this would hold true for the eukaryotic and viral components of the coral microbiome. Utilizing methods such as metagenomics and metatranscriptomics would allow characterization of the whole coral holobiont. It is also important to acknowledge that the pre-exposure sampling of the corals could have contributed to increased mortality or disease signs via the resulting wound. We observed, however, little to no disease or mortality in corals that received the HTS or *SP* indicating that disease manifestation was indeed due to the pathogenic inoculations administered (WBTi DS or *SM*). Our results should still be interpreted conservatively in the context of the relative treatment effects and not as a representation of true transmission rates observed in the field. Future work following similar methodologies should be conducted with a subset of control corals receiving no inoculation after sampling, thereby identifying any potential detrimental effects due the sampling itself.

We identified that the STHS did not elicit a strong response from the coral host, or the prokaryiome, and did not increase disease susceptibility in *A. palmata*. This may therefore make *A. palmata* a better species for restoration practitioners to focus on in the Caribbean rather than *A. cervicornis*. To provide additional strength to this argument a wider range of temperatures should be tested, as well as a wider range of time periods experiencing heat stress, before bleaching signs are reached. Specifically, tailoring the heat stress intensity and duration should be based on historical data for the location of collection of the corals, as well as relevant thresholds that are present in the scientific literature. This will allow identification of potential tipping points before the bleaching threshold is reached, or whether active bleaching signs are this tipping point, and required for a loss in disease resistance. Future work should also look to test predicted and modeled future temperature scenarios which will allow identification of the impacts of disease exposure on the performance of coral species under different climate change

scenarios. Incorporation of a wider range of genets would also be insightful to catch heat tolerance variability that is within the wider *A. palmata* population.

Baseline immune gene expression drove the difference between the four genets. While we hypothesize that this may be important in patterns of disease susceptibility and resistance, future laboratory experiments are needed. Inclusion of a wider range of genets would allow population variability to be captured and allow identification of whether the signal of immune gene expression holds true.

Our work has highlighted that, for *A. palmata*, there are core profiles of the hosts transcriptomic, and prokaryiome, for different stages of the disease response regardless of disease inoculations used. These findings may provide the foundations for identifying health biomarkers at a transcriptomic and prokaryotic level in *A. palmata*, a tool, which could prove valuable to coral restoration practitioners. Additionally, with no strong signal of genet identity identified in the prokaryiome, this may allow development of a species-wide probiotic treatments for *A. palmata*. An important caveat to note here however is that the majority of 16S rRNA work for *A. palmata* has been carried out on nursery-based corals, and thus the core microbiome observed may only be present in these nursery environments. Additional research to identify whether the core microbiome identified in nursery reared *A. palmata* is similar/different to fully wild *A. palmata* colonies should be undertaken, as this would allow identification of whether the prokaryiome does hold true over space and time. Identifying similarities in the prokaryiome between fully wild and nursery corals would also provide management benefits, as probiotic treatments developed from nursery corals could be used in wild outcrops. Finally, our results provide evidence that focusing on a coral's response to infection may be as/more useful than attempting to identify causative pathogen(s) due to the same expression and prokaryotic communities occurring despite different disease inoculations. With the difficulty in identifying causative coral pathogens, due to the marine environment and complex coral holobiont, focusing on these core immune and final state opportunistic prokaryotic communities will allow potential health assays to be developed, active interventions to occur, and potential increased management of coral populations in the wild and nursery systems.

## Supporting information

**S1 Fig. Computer aided design model of disease rack and image of disease inoculation setup in raceway.** A) Computer aided design model of the disease rack setup. Each rack housed eight experimental jars with each jar receiving a dedicated water supply secured via 3D printed clips. Each raceway held five of these racks resulting in 40 experimental jars per raceway. Jar lip was above the raceway water level allowing high replication within one raceway minimizing potential tank effect as well as mitigating cross-contamination between experimental fragments. B) Example of a raceway with five of the disease racks shown in part A). Dedicated 3D printed water clips providing water flow to each jar are shown in red circle for one rack. Experimental *A. palmata* fragments are visible within the experimental jars. (TIF)

**S2 Fig. There were different 16S profiles between the disease inoculations administered to healthy *Acropora palmata* fragments.** A) Principal component [PC] analysis showed separation of the different disease inoculations. PC1 (59%) identified differences between the slurry inoculations (WBTi Disease Slurry [WBTi DS], Healthy Tissue Slurry [HTS]) and the *S. marcescens* [*SM*] inoculations. PC2 (17%) identified a difference between the pathogenic inoculations (WBTi DS and *SM*) and the HTS. B) Shannon-Weiner alpha diversity estimates (y-axis) identified more diverse microbiomes in the WBTi DS inoculations compared to the HTS *and SM* inoculations. X-axis shows the different disease inoculations. C) Relative abundance

analysis of the inoculation samples, at the genus level, with genus with <0.05 abundance removed. Y-axis shows proportions of each genus contributing to relative abundance. Legend to right indicates the fill to genus color for the stacked bar plot. For B) and C): DS1 = WBTi Disease slurry inoculation 1. DS2 = WBTi Disease Slurry inoculation 2. HTS = Healthy Tissue Slurry. *SM1* = *S. marcescens* inoculation one. *SM2* = *S. marcescens* inoculation two.
(TIF)

**S3 Fig. Temperature profiles throughout the experiment and IPAM and buoyant weight results.** A) Temperature data was collected from temperature loggers in the four raceways and ten experimental tanks. Lines plotted are for all the raceways and experimental tanks. The first section (label = "Fragmentation and Recovery) shows the period after fragmentation allowing healing of all wounds. Temperature was set to 27.5˚C. The middle section (labeled = Short-term Heat Stress [STHS]) shows the 15-day temperature stress run in 10 experimental tanks. Ambient corals remained at 27˚C (blue lines). STHS corals experienced a 5-day ramp of 0.5˚C/day, 5-days at 30˚C, and a 5-day ramp back down to 27˚C (0.5˚C/day). The final section (labeled Disease Inoculations) shows temperature of the raceway with disease rack set up. Temperature was set to 27˚C but due to the jar set up within the four raceways, temperature showed larger fluctuation than during the Fragmentation and Recovery period. Temperature fluctuations were roughly +/-1˚C. RW = Raceways (with tanks denoted by numbers 1 to 4), and ET = experimental tanks. STHS tanks are denoted by numbers 1, 9, 10, 12, 15, and red lines. Ambient tanks are denoted by numbers 3, 4, 8, 11, 13, and blue lines. B) Calculated Fv/Fm values for all genets split in ambient (blue line) and STHS (red line). There were no significant differences between ambient or STHS at any of the measurement timepoints (alpha < 0.05). C) Calculated increase in mass for all genet fragments in the ambient (blue line) and STHS (red line) treatments. Percentage change was utilized due to the surface area not being taken of all fragments in the experiment. There were no significant differences between either treatment at any of the measured timepoints. For B) and C), X-axes = Date. Solid vertical lines in plots indicate the start (24[th] July) and end (7[th] August) of the STHS portion of the experiment. Dotted lines indicate the start (29[th] July) and end (3[rd] August) of the 5-days spent at 30˚C for corals experiencing the STHS.
(TIF)

**S4 Fig. Relative risk analysis between; different disease inoculations and genets split by tank treatment, different genets split by disease inoculation, and genets with grouped disease inoculations.** A) Relative risk [RR] analysis of the effect of temperature treatment on risk for each genet within each pathogenic treatment. There were no significant effects of tank temperature on the relative risk for each genet within the Disease Slurry RR (WBTi Disease Slurry [WBTi DS] versus healthy tissue slurry [HTS]) or the *Serratia* RR analysis (*Serratia marcescens* [*SM*] versus *Serratia* placebo [*SP*]). Y axis = genets (CN2, CN4, HS1, ML2) followed by inoculation type; DS = relative risk of WBTi DS vs HTS, SM = relative risk of *SM* vs SP. Blue = median risk of each fragment in ambient temperature treatments. Red = median risk of fragments in the short-term heat stress [STHS] temperature treatment. B) Analysis of the Disease Slurry RR (WBTI DS versus healthy HTS) or *Serratia* RR (*SM* versus SP*)* for each genet. There were no differences in risk between the Disease Slurry RR and *Serratia* RR for genets CN2, CN4, and HS1. There was a significant difference between the relative risks for genet ML2 (alpha < 0.01). Y axes = genets (CN2, CN4, HS1, ML2) with each genet split by: Pink = median risk of each genet for the *Serratia* RR, light green = median risk of each genet for the calculated Disease Slurry RR. C) RR analysis grouping all disease inoculations into either pathogenic (WBTi DS and *SM*) or control (HTS) and SP). There were no significant differences for Pathogenic RR between any of the genets. Y-axis = Genets (CN2, CN4, HS1, ML2)

with genet color identified below plot. For A), B) and C), the Bayesian relative risk analysis, on a log scale, was used for different subsets of the data. Lines depict the 95% credible intervals of the Bayesian analysis. X-axes = log scale. Y-axes = each respective plots variable. 95% confidence intervals that are positive and do not cross 1 indicate a significant higher risk for the pathogenic inoculation than the control inoculation. 95% confidence intervals that are less than 1 indicate a significant lower risk for pathogenic inoculations than the control inoculations. For statistical comparisons between RR please see S1 File.
(TIF)

**S5 Fig. Principal component analysis and differential expression analysis identified a weak effect of the short-term heat stress on transcriptomic coral samples.** A) All coral samples (control, visually healthy and diseased) showed no clear separation of 95% confidence intervals between ambient and short-term heat stressed [STHS] corals. B) Pre-exposure coral samples showed no clear separation of 95% confidence intervals between the STHS and the ambient temperature treatments in PC analysis. Differential expression analysis identified 12 significantly upregulated and 12 significantly downregulated genes. C) Disease inoculated coral samples showed no clear separation of 95% confidence intervals between the STHS and ambient temperature treatments in PC analysis. Differential expression only identified 21 significantly upregulated genes. D) Heatmap showing the 28 significantly differentially expressed genes identified for the pre-exposure coral samples. Heatmap was generated using the variance stabilized transformed [VST] counts of pre-exposure coral samples. Hierarchical clustering of columns (pre-exposure coral samples, dendrogram shown) and rows (genes, dendrogram not shown) identified clustering by temperature treatment with some overlap. E) Heatmap showing the 21 significantly differentially expressed genes identified from the disease inoculated coral samples. Heatmap was generated using the VST counts of disease inoculated coral samples. Hierarchical clustering of columns (disease inoculated coral samples, dendrogram shown) and rows (genes, dendrogram not shown) identifies large overlap between STHS and ambient corals. For A), B), and C), the VST counts with genet variance removed was used. Blue dots = ambient corals (maintained at 27˚C). Red dots = STHS corals (5-days at 30˚C). X-axes show principal component 1. Y-axes show principal component 2. For B) and C), green arrow shows significantly upregulated genes (alpha < 0.01, L2FC <-1) and red arrow shows significantly downregulated genes (alpha < 0.01, L2FC <-1). For A) and C), shapes show visual health status [VHS] of each coral sample.
(TIF)

**S6 Fig. There was a weak effect of temperature treatment on pre-exposure coral samples for the 16s rRNA coral samples, but this was lost after disease inoculation.** A) Principal component [PC] analysis of all coral samples identified no clear separation of 95% confidence intervals due to temperature treatment. B) PC analysis of pre-exposure coral samples identified stronger grouping of ambient compared to short-term heat stress coral samples, with this being significant in PERMANOVA analysis. C) PC analysis identified no clear separation of disease inoculated coral samples due to temperature treatment. D) Correlative analysis between PCs and traits (temperature treatment and genet identity) interest for pre-exposure coral samples. $R^2$ correlation is shown within each heatmap cell with fill showing stronger correlations (green) to weaker correlations (yellow to white). Number of stars within heatmap cells shows the significance of $R^2$ correlations to principal component and metadata variable (* = <0.05, ** = <0.01, *** = <0.001, **** = < 0.0001). E) Shannon-Weiner alpha diversity estimates between ambient and short-term heat stress [STHS] for pre-exposure corals. Points are the Shannon-Weiner alpha metric mean estimates with bars showing calculated confidence intervals. There was a significant difference (alpha < 0.01) between ambient and STHS stress

pre-exposure coral samples in the Shannon-Weiner alpha diversity estimates. F) Relative abundance analysis, to the genus level, between ambient and STHS pre-exposure coral samples. Genera with < 0.01 average abundance were excluded from visualization. Genera bar color fills are shown in legend to right of plot. For A), B), and C), the center logged transformed [CLR] ASV counts were used for each subset of data. Blue dots = ambient corals (maintained at 27˚C). Red dots = STHS corals (5-days at 30˚C). X-axes show principal component 1. Y-axes show principal component 2. For B) and C), green arrow shows significantly increased abundance ASVs (alpha < 0.01, LFC <-1) and red arrow shows significantly decreased abundance ASVs (alpha < 0.01, LFC <-1). For A) and C), shapes show visual health status [VHS] of each coral sample.
(TIF)

**S7 Fig. Transcriptomic WGCNA plots and full correlation to trait heatmap.** A) Scale independence and mean connectivity plots from WGCNA pipeline identifying a soft power of 7 which was used in adjacency matrix calculations. B) Hierarchical clustering of identified module eigengenes with the red horizontal line indicating the cut height (0.25) for merging of modules. C) Cluster dendrogram showing the dynamic tree height with pre-merged modules (Dynamic Tree Cut) and post-merged modules (Merged Dynamic). D) Full module to metadata correlation heatmap for all 16 modules identified through WGCNA analysis. Modules are rows and generic color name is specified to the left of heatmap. Columns are metadata traits split into: a) grouped visual health status (pre-exposure, visually healthy, and diseased), b) visually healthy corals split by disease inoculations (healthy tissue slurry [HTS], WBTi disease slurry [WBTi DS], *Serratia* placebo [SP], and *Serratia marcescens* [SM]), and c) diseased coral samples spilt by disease inoculation (WBTi DS, and SM). Heatmap fill shows positive correlations (red) and negative correlations (blue). For each cell, the upper value identifies module correlation with metadata trait, lower value shows significance of the correlation.
(TIF)

**S8 Fig. 16S rRNA abundance WGCNA plots and full correlation to trait heatmap.** A) Scale independence and mean connectivity plots from WGCNA pipeline identifying a soft power of 12 which was used in adjacency matrix calculations. B) Hierarchical clustering of identified module eigengenes with the red horizontal line indicating the cut height (0.35) for merging of modules. C) Cluster dendrogram showing the dynamic tree height with pre-merged modules (Dynamic Tree Cut) and post-merged modules (Merged Dynamic). D) Full module to metadata correlation heatmap for all 6 modules identified through WGCNA analysis. Modules are rows and generic color name is specified to the left of heatmap. The "Grey" module was removed from this plot. Columns are metadata traits split into a) grouped visual health status [VHS] (pre-exposure, visually healthy, and diseased), b) visually healthy corals split by disease inoculations (healthy tissue slurry [HTS], WBTi disease slurry [WBTi DS], *Serratia* placebo [SP], and *Serratia marcescens* [SM]), and c) diseased coral samples spilt by disease inoculation (WBTi DS, and SM). Heatmap fill show positive correlations (red) and negative correlations (blue). For each cell, upper value identifies module correlation with metadata trait, lower value shows significance of the correlation.
(TIF)

**S9 Fig. 16s rRNA inferred metagenomic function WGCNA plots and full correlation to trait heatmap.** A) Scale independence and mean connectivity plots from WGCNA pipeline identifying a soft power of 8 which was used in adjacency matrix calculations. B) Hierarchical clustering of identified module eigengenes with the red horizontal line indicating the cut height (0.25) for merging of modules. C) Cluster dendrogram showing the dynamic tree height

with pre-merged modules (Dynamic Tree Cut) and post-merged modules (Merged Dynamic). D) Full module to metadata correlation heatmap for the 18 modules identified through WGCNA analysis. Modules are rows and generic color name is specified to the left of heatmap. Columns are metadata traits split into a) grouped visual health status [VHS] (pre-exposure, visually healthy, and diseased), b) visually healthy corals split by disease inoculations (healthy tissue slurry [HTS], WBTi disease slurry [WBTi DS], *Serratia* placebo [SP], and *Serratia marcescens* [SM]), and c) diseased coral samples spilt by disease inoculation (WBTi DS, and *SM*). Heatmap fill show positive correlations (red) and negative correlations (blue). For each cell, top value identifies module correlation with metadata trait, bottom value shows significance of the correlation.
(TIF)

**S10 Fig. Genet identity was the largest driver of transcriptomic variance and was significantly enriched for innate immune genes.** A) Correlation matrix of the genet and visual health status [VHS] variables and their correlations to identified principal components [PC]. $R^2$ correlation is shown within each heatmap cell with fill showing stronger correlations (green) to weaker correlations (yellow to white). Number of stars within heatmap cells shows the significance of $R^2$ correlations to PC and metadata variable (Genet, VHS). Stars signify significance levels, * = <0.05, ** = <0.01, *** = <0.001, **** = < 0.0001. B) Visualization of PC2 and PC3 showed strong clustering by genet identity. Circles are all coral samples (control, visually healthy, and diseased). Ellipsis show the 95% confidence intervals calculated for samples within each genet. C) Left Heatmap shows the 75 most significant genes, identified from the likelihood ratio test [LRT] in DeSeq2, linked to immune GO terms from GO enrichment analysis. Main heatmap was generated using the variance stabilized transformed [VST] counts. Hierarchical clustering of columns (coral samples, dendrogram shown) and rows (genes, dendrogram not shown) was performed. Hierarchical clustering of the columns (coral samples) showed strong clustering of genet identity for genet ML2 (dark grey), CN4 (dark blue), and CN2 (wheat). Genet HS1 (green) also generated a cluster but with inclusion of samples from the other genets. Hierarchical clustering of genes (rows) also identified expression clusters for specific genets. Right heatmap is a presence (black fill) absence (white fill) of genes to the three identified immune GO terms from GO enrichment analysis: *Immune Response* (GO:0006955), *Regulation of Immune Response* (GO:0050776), and *Innate Immune Response* (GO:0045087).
(TIF)

**S11 Fig. There was a weak effect of genet identity identified through principal component correlative analysis and relative abundance analysis for the 16s rRNA analysis.** A) Correlation matrix of principal components [PC] 1 to PC8 to the visual health status [VHS] and genet identity metadata variables. Correlation of metadata trait to PC is shown in each cell, with fill showing stronger correlations (green) to weaker correlations (yellow to white). Number of stars within heatmap cells shows the significance of $R^2$ correlations to principal component and metadata variable (* = <0.05, ** = <0.01, *** = <0.001, **** = < 0.0001). B) Visualization of PC4 (3% variance) and PC5 (2% variance) due to significant correlations of genet through correlative analysis. Samples are colored only by genet with legend to the right of plot. C) Relative abundance analysis for each genet split by VHS and disease inoculations. Genera with <0.025 average abundance were excluded from visualization. Top left plot = CN2, top right plot = CN4, bottom left plot = HS1, bottom right plot = ML2. For each genet plot, x-axis shows: PE (pre-exposure), visually healthy coral samples split by disease inoculation (HTS = healthy tissue slurry, WBTi DS = white band type I disease slurry, SP = *Serratia* placebo, SM = *Serratia marcescens*), and diseased corals split by disease inoculations (WBTi DS = white band type I disease slurry, SM = *Serratia marcescens*). For each pot y-axes show

relative proportion. For all plots, bar fills follow the legend to the right showing genus to color fill.
(TIF)

**S1 File. Supplementary methods for IPAM and buoyant weight, relative risk analysis, 16S library preparation, transcriptomic pre-processing bioinformatic pipeline, 16S rRNA pre-processing bioinformatic pipeline.**
(DOCX)

**S2 File. Identified ASVs used for analysis, mitochondrial chloroplast, and eukaryotic ASVs removed, and disease inoculation ASVs retained for analysis.**
(XLSX)

**S3 File. Transcriptomic differential expression analysis between short-term heat stress and ambient corals.**
(XLSX)

**S4 File. 16S rRNA differential abundance analysis between short-term heat stress and ambient corals.**
(XLSX)

**S5 File. Transcriptomic co-expression hub genes, module gene lists, and overlap with Young *et al* (2020).**
(XLSX)

**S6 File. Co-expression GO results and genes to GO terms.**
(XLSX)

**S7 File. Co-expression KEGG results and gene to KEGG terms.**
(XLSX)

**S8 File. Relative abundance percentages between visual health states.**
(CSV)

**S9 File. 16S rRNA co-abundance hub ASVs and module ASV lists.**
(XLSX)

**S10 File. 16S rRNA inferred metagenomic function co-expression hub genes, module gene lists, and KEGG enrichment results.**
(XLSX)

**S11 File. Transcriptomic genet LRT results and enrichment analyses.**
(XLSX)

**S12 File. Pre-exposure core microbiome and relative abundance analyses between the four genets.**
(XLSX)

## Acknowledgments

We would like to thank Coral Restoration Foundation for donating *Acropora palmata* fragments for this experiment, as well as Dr. Diego Lirman and the Rescue a Reef Lab, University of Miami, for providing healthy and diseased fragments of *Acropora cervicornis*. We would also like to thank Kathryn Sutherland for supplying the PDR60 strain of *Serratia marcescens*, and Dr. Javier del Campo's lab, University of Miami, for helping maintain the culture. Thank

you also to Dr. Iliana Baum and Dr. Sheila Kitchen for providing version 3 of the *Acropora palmata* genome. Finally, we thank the reviewers for their time and effort in providing comments and reviewing our article.

## Author Contributions

**Conceptualization:** Benjamin D. Young, Stephanie M. Rosales, Ian C. Enochs, Nikki Traylor-Knowles.

**Data curation:** Benjamin D. Young.

**Formal analysis:** Benjamin D. Young, Stephanie M. Rosales, Gabrielle L. D'Alonso.

**Funding acquisition:** Stephanie M. Rosales, Ian C. Enochs, Nikki Traylor-Knowles.

**Investigation:** Benjamin D. Young, Gabrielle L. D'Alonso.

**Methodology:** Benjamin D. Young, Stephanie M. Rosales, Ian C. Enochs, Graham Kolodziej, Nathan Formel, Nikki Traylor-Knowles.

**Project administration:** Benjamin D. Young, Nikki Traylor-Knowles.

**Resources:** Ian C. Enochs, Graham Kolodziej, Nathan Formel, Amelia Moura.

**Software:** Benjamin D. Young.

**Supervision:** Stephanie M. Rosales, Ian C. Enochs, Nikki Traylor-Knowles.

**Visualization:** Benjamin D. Young.

**Writing – original draft:** Benjamin D. Young.

**Writing – review & editing:** Benjamin D. Young, Stephanie M. Rosales, Ian C. Enochs, Graham Kolodziej, Nathan Formel, Amelia Moura, Gabrielle L. D'Alonso, Nikki Traylor-Knowles.

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
