## [Decision Letter · Decision Letter 0]

12 Mar 2023

PONE-D-23-02218Different disease inoculations cause common responses of the host immune system and prokaryotic component of the microbiome in Acropora palmataPLOS ONE

Dear Dr. Young,

Thank you for submitting your manuscript to PLOS ONE. After careful consideration, we feel that it has merit but does not fully meet PLOS ONE’s publication criteria as it currently stands. Therefore, we invite you to submit a revised version of the manuscript that addresses the points raised during the review process.

I would recommend the authors to provide a careful and concise revision of the manuscript and also attract their attention to generate figures as per PloS one publication standard. The abstract needs a thorough readout and minor rewrite to make it clearer to the broader audience. The other issues raised by the reviewer 2 sounds reasonable to me. Please submit your revised manuscript by Apr 26 2023 11:59PM. If you will need more time than this to complete your revisions, please reply to this message or contact the journal office at plosone@plos.org. Please include the following items when submitting your revised manuscript:A rebuttal letter that responds to each point raised by the academic editor and reviewer(s). You should upload this letter as a separate file labeled 'Response to Reviewers'.A marked-up copy of your manuscript that highlights changes made to the original version. You should upload this as a separate file labeled 'Revised Manuscript with Track Changes'.An unmarked version of your revised paper without tracked changes. You should upload this as a separate file labeled 'Manuscript'.If applicable, we recommend that you deposit your laboratory protocols in protocols.io to enhance the reproducibility of your results. Protocols.io assigns your protocol its own identifier (DOI) so that it can be cited independently in the future. For instructions see: https://journals.plos.org/plosone/s/submission-guidelines#loc-laboratory-protocols. Additionally, PLOS ONE offers an option for publishing peer-reviewed Lab Protocol articles, which describe protocols hosted on protocols.io. Read more information on sharing protocols at https://plos.org/protocols?utm_medium=editorial-email&utm_source=authorletters&utm_campaign=protocols.

We look forward to receiving your revised manuscript.

Kind regards,

Tarunendu Mapder, Ph.D.

Academic Editor

PLOS ONE

Journal Requirements:

   "ICE, Grant number 31252, "NOAA Coral Reef Conservation Program. ICE provided experimental design expertise, lab resources, and manuscript comments and edits. NTK supported by NSF Grant 1951826 and Protect our Reefs Grant 2018-23. NTK helped plan the experiment, and provided direction on bioinformatic analysis, as well as manuscript comments and edits on drafts. "  

4. Please ensure that you include a title page within your main document. You should list all authors and all affiliations as per our author instructions and clearly indicate the corresponding author.

Reviewers' comments:

Reviewer's Responses to Questions

**Comments to the Author**

1. Is the manuscript technically sound, and do the data support the conclusions?

Reviewer #1: Yes

Reviewer #2: Yes

2. Has the statistical analysis been performed appropriately and rigorously? 

Reviewer #1: Yes

Reviewer #2: Yes

3. Have the authors made all data underlying the findings in their manuscript fully available?

Reviewer #1: Yes

Reviewer #2: Yes

4. Is the manuscript presented in an intelligible fashion and written in standard English?

Reviewer #1: Yes

Reviewer #2: Yes

5. Review Comments to the Author

Reviewer #1: Thank you for submitting this good work to Plos one. The paper is well written, and the data analysis is sound and sufficient. I do not have additional comments except that the resolution of Figs in the system was a bit low.

Reviewer #2: The manuscript entitled “Different disease inoculations cause common responses of the host immune system and prokaryotic component of the microbiome in Acropora palmata” represents a thorough and needed investigation of host transcriptomic and microbiome responses to distinct disease challenges in a critically important reef-building species. I find the main strength of the paper to be the evidence and discussion of microbiome succession during early stages of coral disease. Most coral disease studies focus entirely on corals that already have advanced signs of disease and have very likely progressed past the microbial determinants and host responses to the initial causative event. My main criticism with the paper involves the heat stress component, which is ill-defined in the context of environmental variation for these corals and the results of which are confusingly reported. Specific recommendations for this limitation, as well as other recommendations to improve clarity and better inform readers of methodologies used, are described below. Lastly, I found the manuscript to be rather overwhelming in length and supplements. I appreciate the thorough analysis and transparent reporting, but I encourage the authors to take advantage of any opportunities to minimize redundancy and be more concise.

Methods:

Line 125: Please specify how many ramets of each genet. Alternatively, this information could be included in the tank descriptions.

Line 138: How do the 27C control and 30C heat stress compare to minimum-average-maximum temperatures these corals experience in the nursery in these same months? Relative to seasonal variation and SSTs during known bleaching events, is 30C a light, moderate, or severe heat stress?

Line 153: Did the authors prepare WBTi DS twice (once for each inoculation) or did they preserve it between inoculations (or something else)? Please clarify.

Line 165: The same healthy fragments were used to generate HTS: does this mean that some slurry was removed from a ramet one day and then more tissue was removed another day? Or does this mean that slurry was removed from a ramet and then more tissue was removed from another ramet of the same genet for the next inoculation? Or does this mean that tissue was removed once and preserved between inoculation? Or something else? Please clarify.

Line 174: Was this process repeated for the second inoculation? If so, how did the authors ensure consistency in inoculation loads between doses (e.g., plating out/measuring OD/calculating CFU/mL?) Please clarify.

Line 189: Explain the coral sampling procedure (e.g., approximate size of tissue removed), including how any lesion produced by the sampling may have affected coral health in subsequent exposures.

Line 240: Please clarify the basis of enrichment. Only significant genes were used as input, and this was run against the total list of genes? Or was the enrichment based on p-value within the list of significant genes?

Line 251: Similar as above, please clarify the basis of enrichment. For example, within-module genes were input run against all other genes? Or was some continuous value (e.g., kME?) used as the basis of enrichment.

Line 310: The module-on-module correlation described here is complicated and could use another sentence or two for clarification. If I’m understanding correctly, each sample has a continuous eigengene value for each module, and those values are correlated? I think recall per-sample MEs but the generic WGCNA description of ME as the first PC of the module lends the impression of a single ME per module, which wouldn’t facilitate correlation analysis.

Line 361: I’m not sure how to interpret this sentence. “…when splitting these six modules by the four disease inoculations” – does this mean when looking at the correlations of each module with each of the four disease inoculations? When I do that, I see that the directionality of correlation is the same across disease inoculations for these six modules (where shown). But then “all correlations for visually healthy and diseased coral samples showed the same correlation directionality” isn’t true for these six modules. Blue, Black, and Dark Green have opposite signs between VH and D. Please clarify.

Results:

Line 322: The methods only describe alignment to the genome. Why is the alignment rate to the transcriptome mentioned here? Are the alignments to the transcriptome used in the analysis?

Line 356: I think “2A” and “2B” are switched in this sentence. Figure 2A shows groupings by inoculation and Figure 2B shows groupings by visual status. Also, both appear to show substantial separation by inoculation or visual status. Please provide statistics to support the statement that analysis of variance did not find significant differences between disease inoculations. It would be helpful if those stats could be provided on the figures directly as well.

Line 429: Was any manual characterization of this predicted/uncharacterized gene attempted? E.g., manual BLAST to check for updated annotations or protein domain prediction.

Line 449: “regardless of which disease inoculations was administered (Fig 3A)” but Fig 3A shows some separation grouping by health status. Please provide stats.

Line 450: “with no unique profiles identified due to the four disease inoculations (Fig 3B)”, but again, Fig 3B does seem to show some clustering. Please provide ANOVA or similar stats for these analyses.

Line 458: The Shannon value stated in the text (2.25) doesn’t match the value shown in 3C (looks very similar to the VH value). The 2.25 value would be significantly less than the 4.52 value, but the values shown on the figure look similar. Please double-check stats here and make sure you’re reporting consistent values.

Line 572: The description of this section only references transcriptomic and 16S results, but it also includes all the heat stress results, disease outcomes, buoyant weight, and PAM data. I recommend restructuring for clarity.

Line 605: I think the heat stress results could be moved sooner. I’m guessing the lack of temperature effect is why the earlier sections of the results didn’t include any analysis of temperature. It’s unclear why these results are included in the current section, and up until this point, I had been wondering what happened to heat treatment and what the differences in disease outcome were between groups.

Figures:

Figure 2: A and B are switched.

Figure 5: The simplified layout of the main correlative heatmap didn’t add anything for me. I spent a few moments trying to figure out why those boxes were empty before I realized what was happening. I think the layout of the main heatmap is fine, given that it has the separation between the left and right parts with descriptive labels above them. One could simply add the words “modules” to the top labels and a “Transcriptomic modules” (maybe “Coral transcriptomic modules” to be extra clear) label along the y-axis to completely recapitulate the “simplified” diagram.

S6: This figure could be a good place to include some sort of seasonal data trends layered on top of tank measurements. E.g., average min/max values for July/August over the last few years as additional lines on S6A.

Discussion:

Line 824: This caveat is important and would be a good place to reference normal seasonal values for this species in situ. Has paling or any other signs of stress been observed for this species at 30C for ~15 days? How normal is it for them to experience this level of temperature stress?

Line 826: What was the duration and magnitude of short-term heat stress in this cited study that did find clear transcriptomic responses to STHS? How does that compare to the parameters in this study?

Line 835: One can theoretically agree with the premise of this section (that differences in baseline immune expression drive differences in susceptibility), but this study doesn’t provide any supporting evidence. You observed the differences in immune gene expression, but not the differences in susceptibility (S7C). The last sentence (Line 847–848) is not supported. Please rephrase or clarify the connection between this hypothesis and the observations of this study. E.g., why didn’t you see that genets with higher immune expression having lower rates of disease?

Line 889: I don’t understand this line: “inclusion of experimental corals not exhibiting any inoculations.” I think this is a suggestion to include control (unchallenged) corals in experiments, but I’m not sure given the “exhibiting inoculations” language. Please clarify.

Line 901: By including observational temperature data, one could make a more informed recommendation here than simply “wider range of temperatures.” For example, the recommendation could be to design experiments around observed and predicted temperature models for these species/populations.

Line 915: Do the authors expect to find a similar lack of a “strong signal of genet identity” in prokaryiome communities in wild A. palmata? I could imagine an expectation of more similar microbiota among members of a restoration population than among “rare outcrops” unless the authors have additional citations to support the core prokaryiome across this species’ range. If the authors agree, maybe add a clarifying statement here or in earlier discussions of the “core prokaryiome” to address the possibility that this finding is specific to nursery corals.

Minor comments:

No action needed: I’m just curious if the authors are inventing the term “prokaryiome” here. I’ve never heard it and a quick google search yielded nothing. I personally would accept an original description of the “bacterial and archaeal microbiome” and then subsequent use of simply “microbiome.” I know a small but vocal contingent of microbiologists who are trying to retire the word “prokaryote” in favor of “bacteria” or “archaea” where appropriate, and who will probably make fussy noises at this new term. But I also appreciate that this study describes a situation where archaea and bacteria are described together with no implication of their evolutionary relationship, and therefore this use seems appropriate for “prokaryote” to distinguish the two groups as “non-eukaryotic.”

Very minor grammatical comment that I only mention because it considerably impedes understanding in some cases: The authors frequently use “this” without a clear antecedent. For example, what does “this” reference in Line 46? Prokaryotic and eukaryotic life or marine environments or infectious diseases? None of the available antecedents from the prior line seem to fit the description of something that “includes healthy ecosystems” in the way the next line describes. Lines 58 (this prevalence of disease?), 60 (this/these criteria?), 63 (this failure to confidently describe etiology?), 67 (this complexity?) and others are less ambiguous, but I’m still not certain I’m guessing the correct “this” in every case.

Line 311: “run-on” can be “run on”

Line 800: “omics” was previously stylized as “’ ’omics”

6. PLOS authors have the option to publish the peer review history of their article (what does this mean?). If published, this will include your full peer review and any attached files.

Reviewer #1: No

Reviewer #2: No

---

## [Author Response · Author response to Decision Letter 0]

11 Apr 2023

Good afternoon 

Please find the revised manuscript following the comments from the editor and the two reviewers. We are very thankful for the time and effort put in by the reviewers, and have incorporated the suggestions and requests that they asked for. All responses to editor and reviewer comments are in the uploaded word document response_to_reviewers.docx, with tracked changes in the tracked change word document. Please let me know if you require any more information, and I will provide it asap. 

Dr. Young

---

## [Editor Report · Decision Letter 1]

15 May 2023

Different disease inoculations cause common responses of the host immune system and prokaryotic component of the microbiome in Acropora palmata

PONE-D-23-02218R1

Dear Dr. Young,

We’re pleased to inform you that your manuscript has been judged scientifically suitable for publication and will be formally accepted for publication once it meets all outstanding technical requirements.

Kind regards,

Tarunendu Mapder, Ph.D.

Academic Editor

PLOS ONE

---

## [Editor Report · Acceptance letter]

18 May 2023

PONE-D-23-02218R1 

Different disease inoculations cause common responses of the host immune system and prokaryotic component of the microbiome in *Acropora palmata*

Dear Dr. Young:

I'm pleased to inform you that your manuscript has been deemed suitable for publication in PLOS ONE. Congratulations! Your manuscript is now with our production department. 

Kind regards, 

on behalf of

Dr. Tarunendu Mapder 

Academic Editor

PLOS ONE